# AI-identified CD133-targeting natural compounds demonstrate differential anti-tumor effects and mechanisms in pan-cancer models

Yibo Hou [ID][1,4], Zixian Wang[1,4], Wenlin Wang[2,4], Qing Tang[2], Yongde Cai[3], Siyang Yu[1], Jin Wang [ID][3], Xiu Yan[3], Guocai Wang[2], Peter E Lobie[1], Yubo Zhang[2✉], Xiaoyong Dai[2✉] & Shaohua Ma [ID][1✉]

## Abstract

**Advanced algorithms have significantly improved the efficiency of in vitro screening for protein-interactive compounds. However, target antigen (TAA/TSA)-based drug discovery remains challenging, as predictions of compound-protein interaction (CPI) based solely on molecular structure fail to fully elucidate the underlying mechanisms. In this study, we utilized deep learning, specifically TransformerCPI to screen active molecules from a Chinese herb compound library based on protein sequences. Two natural products, Polyphyllin V and Polyphyllin H, were identified as targeting the pan-cancer marker CD133. Their anti-tumor efficacy and safety were confirmed across validation in cancer cell lines, tumor patient-derived organoids, and animal models. Despite their analogous structures and binding affinity to CD133, Polyphyllin V suppresses the PI3K-AKT pathway, inducing pyroptosis and blockage of mitophagy, whereas Polyphyllin H inhibits the Wnt/β-catenin pathway and triggers apoptosis. These distinct mechanisms underscore the potential of combining AI-driven screening with biological validation. This AI-to-patient pipeline identifies Polyphyllin V and Polyphyllin H as CD133-targeted drugs for pan-cancer therapy, and reveals the limitations of virtual screening alone and emphasizes the necessity of live model evaluation in AI-based therapeutic discovery.**

**Keywords** CD133; TransformerCPI; Natural Products; Mitophagy; Apoptosis
**Subject Categories** Cancer; Computational Biology; Pharmacology & Drug Discovery

## Introduction

The prevalence of advanced-stage cancer dramatically impacts patient survival and quality of life. The standard of care typically involves a convergence of surgical intervention, chemotherapeutic regimens, and/or radiation therapy (Crosby et al, 2022; Bray et al, 2024; Passaro et al, 2024). Targeted therapeutic approaches, such as antibodies against tumor-associated/specific antigens (TAA/TSA) and small-molecule inhibitors targeting critical signaling pathways, have been implemented based on antigen expression and patient mutation profiles (Hurwitz et al, 2004; Andre et al, 2022). However, these approaches face challenges such as a shortage of inhibitors, off-target effects, significant toxicity, and drug resistance, underscoring the need for novel therapeutic strategies.

CD133 is a crucial glycoprotein with pentaspan-transmembrane structure involved in identification of diverse kinds of normal stem cells and various cancer stem-like cells (CSC) (Li 2013; Sansone et al, 2016; Mancebo et al, 2017; Zhou et al, 2021; Yamashita et al, 2022; Ahn et al, 2024), including colorectal cancer (CRC) (Singh et al, 2004; Ricci-Vitiani et al, 2007; Calvanese et al, 2022), liver cancer (Stephanie 2013) and lung cancer (Wu et al, 2014). CD133-high expression cells are positively correlated with chemoresistance. As a CSC marker, CD133 overexpression is linked to aggressive tumor phenotypes and poor clinical outcomes (Ren et al, 2013). Research indicates that tumor stem cells with CD44 + /α2β1hi/ CD133+ phenotypes, derived from 40 patient biopsy samples, exhibit 3.7 times the self-renewal capability of CD133-negative cells (Vander Griend et al, 2008). However, therapeutics targeting CD133 remain in the early stages of development. The complex pentaspan transmembrane structure of CD133 and its heterogeneous expression in tumors hinder binding and the achievement of the desired therapeutic effect. In addition, the presence of multiple epitopes with varying glycosylation patterns complicates the development of antibody-based drugs (Glumac and LeBeau 2018; Pleskač et al, 2024). Therefore, there is an urgent need for efficient screening implementation to find CD133 target drugs.

Deep-learning technology is rapidly accelerating drug discovery and the study of compound-protein interaction (CPI) (Liu et al, 2023; Mou et al, 2023; Mullowney et al, 2023; Zhang et al, 2025). Identifying CPIs is crucial in drug discovery and chemical genomics, and can be achieved by analyzing protein 3D structures or sequences when 3D information is unavailable (Sadybekov and

[1]Institute of Biopharmaceutical and Health Engineering, Tsinghua Shenzhen International Graduate School (SIGS), Tsinghua University, Shenzhen 518055, China. [2]Department of Physiology, School of Medicine; Institute of Traditional Chinese Medicine & Natural Products, College of Pharmacy, and Guangdong Province Key Laboratory of Pharmacodynamic Constituents of TCM and New Drugs Research, Jinan University, Guangzhou 510632, China. [3]Synorg Biotechnology (Shenzhen) Co. Ltd., Shenzhen 518107, China. [4]These authors contributed equally: Yibo Hou, Zixian Wang, Wenlin Wang. ✉E-mail: ybzhang99@jnu.edu.cn; daixy18@jnu.edu.cn; ma.shaohua@sz.tsinghua.edu.cn

Katritch 2023; Miller et al, 2024). TransformerCPI converts protein and atomic representations into model input, generating predicted values for interactions between target proteins and drug molecules, including interaction strength and solubility (Chen et al, 2020). It can provide a solution for conducting high-throughput screening of protein targets with complex properties, such as CD133.

Natural product molecules have long been integral to pharmacotherapy and represent a promising reservoir of bioactive compounds with diverse chemical structures and pharmacological activities for targeted drug discovery (Ghosh et al, 2024). Natural products typically exhibit more favorable safety profiles and lower toxicity compared to synthetic molecules (Over et al, 2013; Atanasov et al, 2021; Dong and Lei, 2024). However, traditional drug screening is time-consuming and labor-intensive, posing challenges in identifying drugs with specific targeting capabilities (Beniddir et al, 2021). The efficient discovery of natural product leads represents a compelling strategy to overcome current limitations in therapeutic development and improve patient outcomes.

Multiple steroidal saponins extracted from *Paris polyphylla* have been reported to possess anti-tumor (Yin et al, 2014; Quan et al, 2022), anti-inflammatory, antioxidant, hemostatic, and analgesic effects (Zhu et al, 2011; Zeng et al, 2020; Ming-Ming et al, 2021). This study identified and characterized two natural product molecules, Polyphyllin V (PP10) and Polyphyllin H (PP24), which target CD133 in multiple cancer cell types. To date, no study has elucidated the anti-tumor targets and cytotoxic mechanisms of PP10 and PP24. Within this research, to explore the anti-tumor mechanisms, we used a combination of Transformer-based CPI screening and multi-platform validation across cancer cell lines, xenograft mice, and microfluidics-engineered cancer organoid models.

Interestingly, despite analogous structure and binding affinity to CD133, PP10, and PP24 affect downstream signaling pathways differently, leading to distinct anti-tumor mechanisms, as evaluated through multi-platform validation. PP10 suppresses the PI3K–AKT pathway, triggering mitophagy blockage and pyroptosis to eliminate CRC cells. In contrast, PP24 suppresses the Wnt/β-catenin pathway and induces apoptosis, hence reducing CRC proliferation. In this study, we highlight the importance of using live model evaluation to compensate for the limitations, though of the enormous potential, of AI-based therapeutics discovery.

# Results

## CD133 expression across multiple cancers

To elucidate the impact of CD133 (PROM1) on tumor progression, bioinformatics analyses were performed to assess its expression levels in pan-cancer. Elevated CD133 expression was observed in several cancers in the TCGA dataset, including colon adenocarcinoma (COAD), cholangiocarcinoma (CHOL), pancreatic adenocarcinoma (PAAD), rectum adenocarcinoma (READ), and stomach adenocarcinoma (STAD) (Fig. 1A). In addition, transcriptomic datasets of the TCGA and GEO (GSE21815) demonstrated the expression of CD133 was significantly elevated in CRC patient compared with normal tissues (Fig. 1B,C) (Barrett et al, 2012; TCGA 2024). Furthermore, increased CD133 expression was positively associated with poor prognosis, as confirmed by the

GSE30378 dataset (Fig. 1D,E). Immunohistochemical staining of clinical cancer tissues further confirmed higher CD133 expression in liver, lung, colorectal, and gastric cancers, contrasted with adjacent tissues (Figs. 1F and EV1A). In addition, our results revealed significantly higher CD133 levels of CRC tissues versus adjacent tissues (Fig. 1G,H). As a result, CRC was selected as the primary tumor type for validating the screened targeting compounds.

## Identification of CD133-targeted compounds via Transformer-based CPI screening

Our group extracted and isolated 1123 compounds from various Chinese herbs, including *Garcinia oblongifolia*, *Croton crassifolius*, and *Paris polyphylla* et al, and created a natural product library for AI-based drug screening (Dai et al, 2022; Wu et al, 2022). The Transformer-based algorithm was used to predict the binding potential between CD133 and compounds, aiming to identify potential antagonists. The amino acid sequence of CD133 was input and encoded using the Byte Pair Encoding (BPE) algorithm, breaking it into overlapping trigrams (Zhang et al, 2022). Simultaneously, the SMILES representation of the molecules was input, and features were extracted via RDKit (Fig. 2A–C). The model utilized a multi-head self-attention mechanism to learn features and generate predictions through a feed-forward neural network. This process identified 29 compounds with potential interactions with CD133. The top 6 compounds were selected for further validation of their anti-tumor activity using MTT assays. Among these, PP10 and PP24 exhibited significant cytotoxicity, with the previously studied compound CL6 (polyphyllin II) (Wang et al, 2019; Jin et al, 2024) also showing notable activity (Fig. 2D). However, the difficulty in obtaining and the potential hepatotoxicity seriously hindered the investigation of CL6, it was excluded from the research. In CD133$^{+/+}$ cancer cell lines, PP10 and PP24 showed the greatest cytotoxic effects on CRC cell. The IC$_{50}$ values of PP10 in HCT116 and DLD1 cells were 1.747 μM and 1.563 μM, respectively, while those of PP24 were 0.3154 μM and 0.9958 μM, respectively (Appendix Fig. S1A,B). Therefore, we focused on the anti-tumor effects of PP10 and PP24 in CRC cell lines.

## Affinity verification of PP10 and PP24 to CD133

Molecular docking results revealed that PP10 bound to CD133 through hydrophobic interactions at residues ALA-2, LEU-5, PHE-62, TRP-786, PHE-787, and LYS-791, and formed hydrogen bonds at LEU-3 and PHE-795 (Fig. 2E). Similarly, PP24 bound to CD133 via hydrophobic interactions at LEU-76, LEU-80, LEU-505, ILE-506, and PHE-516, and formed hydrogen bonds at PRO-72 and GLU-97 (Fig. 2F). Surface plasmon resonance (SPR) experiments quantified the binding affinities between CD133 and PP10, as well as PP24, confirming the docking predictions. The response values for PP10 and PP24 binding to CD133 ranged from 0 to 135 RU and 0 to 124 RU, respectively, elevating in a concentration-dependent manner. It indicated strong binding, with dissociation constants (KD) of $5.4 \times 10^{-5}$ M for PP10 and $1.0 \times 10^{-5}$ M for PP24 (Fig. 2G,H). The cellular thermal shift assay (CETSA) further confirmed the drug-protein interactions (Molina et al, 2013; Jafari et al, 2014; Rai et al, 2017), showing increased thermostability and accumulation of CD133 in CRC cell lines treated with either PP10 or PP24 compared to control groups (Fig. 2I,J; Appendix

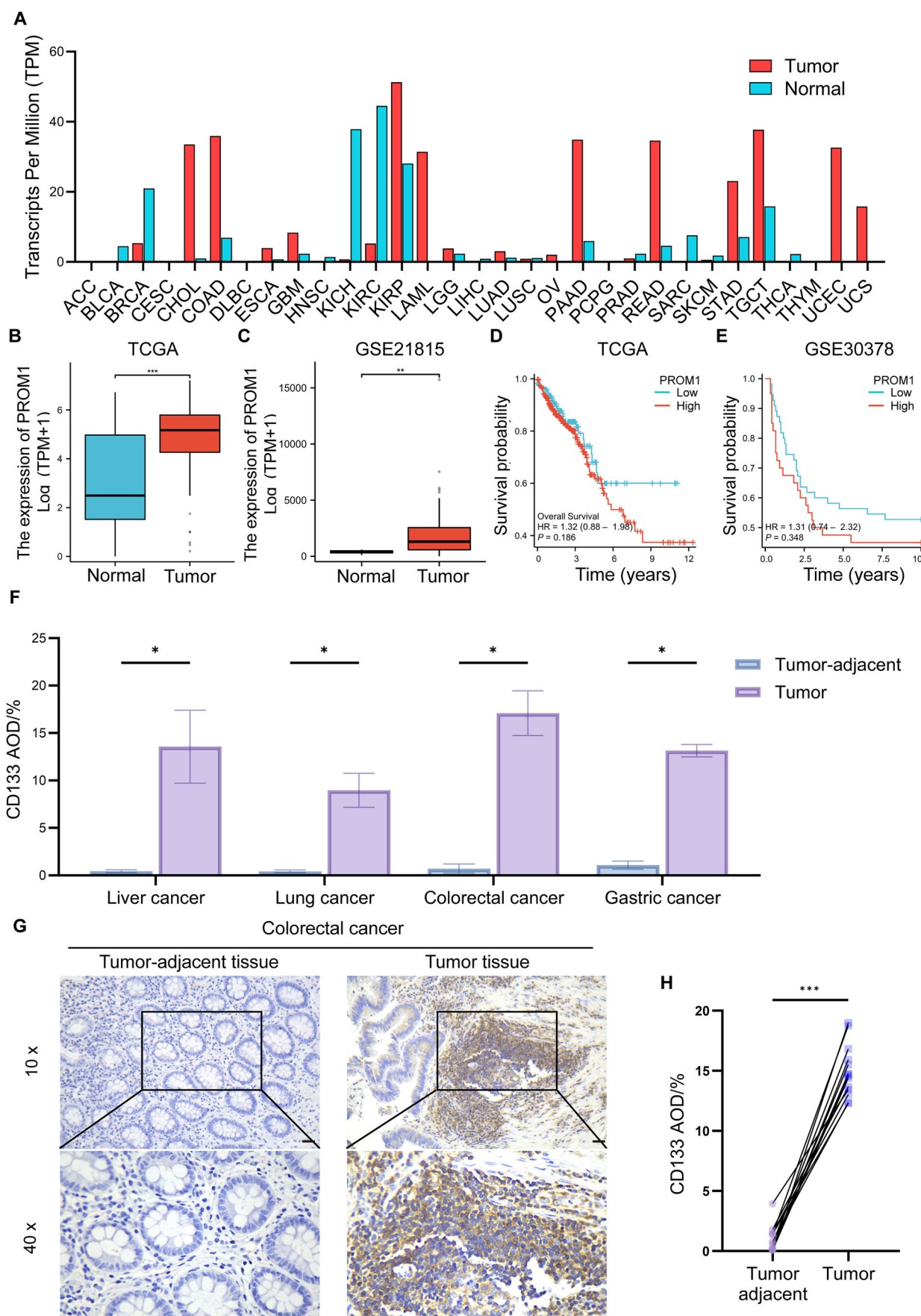

**Figure 1. Expression level of CD133 in CRC patients and prognosis analysis.**

(A) Expression level analysis of CD133 in pan-cancer from the TCGA database. Expression-level analysis of CD133 in CRC from the TCGA database (B) and GSE21815 dataset from the GEO database (C). Kaplan–Meier estimator of overall survival rate based on CD133 expression from the TCGA database (D) and GSE30378 dataset from the GEO database (E). (F) Quantitative IHC results for CD133 expressed in pan-cancer patients as average optical density (AOD) values. (G) IHC staining of CRC patient's tumor tissue section and normal tissue section on CD133. (H) Quantitative IHC results for CD133 expressed in CRC patients as AOD values. Scale bar = 50 µm. Student's *t* test. Exact *P* values are presented in Appendix Table S1. Data are presented as mean ± SD. *$P < 0.05$, **$P < 0.01$, and ***$P < 0.001$. Source data are available online for this figure.

Fig. S1C,D). These findings confirmed the strong binding affinity of PP10 and PP24 to CD133. We constructed a CD133-knockdown HCT116 cell line and used the FHC normal colorectal cell line to assess PP10 and PP24 cytotoxicity on CD133-negative cancer cells and normal cells (Fig. 2K). The $IC_{50}$ values for both cell lines rose from ~1 µM to 18.99–28.64 µM. Specifically, PP10 had an $IC_{50}$ of 21.78 µM in FHC cells and 28.64 µM in shCD113-HCT116 cells, while PP24 showed $IC_{50}$ values of 18.99 µM and 25.39 µM in the respective cell lines (Figs. 2L and EV1D,E). These results show PP10 and PP24 selectively target CD133-overexpressing cancer cells, with lower cytotoxicity in CD133-negative and normal cells.

## PP10 and PP24 inhibit the migration and metastasis of CRC cells

As a CSC marker, CD133 is essential for cancer metastasis and recurrence (Liou 2019; Zhao et al, 2022). The anti-tumor effects of PP10 and PP24 on CRC cells' migration and invasion were investigated. The scratch wound healing assay revealed that in HCT116 cells, the wound in the control group nearly closed after 48 h, while wound healing was significantly inhibited in cells treated with PP10 or PP24, with PP10 showing a stronger inhibitory effect. A similar trend was observed in DLD1 cells as well (Figs. EV2A–C and 3A–C). The migration ability was further assessed utilizing the Transwell assay. Treatment with PP10 or PP24 significantly reduced the number of migration cells as the concentration increased (Figs. EV2D,E and 3D,E). To simulate the extracellular matrix and evaluate invasion, a Matrigel-based assay was conducted, showing significant inhibition of metastasis by PP10 or PP24 (Figs. EV2F,G and 3F,G). A colony formation assay was performed to evaluate the tumorigenicity of CRC cells in vitro after treatment with PP10 or PP24. After 14 days, both the number and size of colonies formed by CRC cells were significantly reduced in a dose-dependent manner (Figs. EV2H,I and 3H,I). Based on the observed inhibition of migration and metastasis by PP10 and PP24, the levels of epithelial-mesenchymal transition (EMT) markers were detected to investigate the underlying mechanisms (Mao et al, 2024). Western blot analysis revealed a slight rise of E-cadherin, while a significant reduction of N-cadherin, Vimentin, and Snail expression for CRC cells treated with PP10 or PP24 (Figs. EV2J,K and 3J; Appendix Fig. S2A–C). These findings suggested that PP10 and PP24 suppressed tumorigenicity, migration, and invasion in CRC cells via inhibiting the EMT pathway.

## PP10 and PP24 inhibit CRC growth in a cell-derived xenograft model with low toxicity

The anti-tumor function of PP10 and PP24 was subsequently evaluated in animal models. HCT116 cells were selected for model

establishment due to their widespread use, and subcutaneous injections were administered into BALB/c nude mice (Fig. 3A). Mice were separated into four groups at random for each Polyphyllin compound treatment. Tumor volume and weight showed that tumors in the PP10 or PP24-treated groups were remarkably smaller compared with the control group (Fig. 3B,C). In addition, tumor volume and tumor weight were decreased in a dose-dependent manner as drug concentration increased (Fig. 3D,E). Interestingly, PP10 and PP24 exerted better anti-tumor effects in the high-concentration group compared to oxaliplatin, a standard CRC clinical drug. No significant change in mouse weight was observed in response to treatment with PP10 or PP24, suggesting that these compounds exhibited low toxicity in vivo (Fig. 3D,E). To further investigate apoptosis in the CDX model, a TUNEL assay was conducted to measure the apoptotic activity by PP10/PP24 (Hou et al, 2021). Staining of tumor tissue sections indicated a notable increase in fluorescence intensity in both PP10 and PP24 treatment groups compared to the control group, indicating effective induction of apoptosis in vivo (Fig. 3F,G; Appendix Fig. S2D,E). In addition to their efficacy against tumors, the impact of Polyphyllin compounds on metabolism and physiological functions was also evaluated. HE staining of organ slides was performed for safety assessment, revealing no significant toxicity and normal tissue morphology (Appendix Fig. S3A,B). Serum analysis of liver and kidney function showed no significant differences across indicators both in PP10/PP24-treated and control groups (Appendix Fig. S3C,D). No adverse effects, such as vomiting, diarrhea, or significant weight loss, were observed during the experiment. In conclusion, PP10 and PP24 demonstrated strong inhibition of CRC growth in the CDX model, coupled with low toxicity and high biosafety.

## PP10 exerts anti-tumor effects via inhibiting PI3K/AKT/mTOR pathway

Total mRNA was extracted from PP10-treated CRC cells and sequenced to explore the impact of PP10 on the downstream pathway of CD133. Significantly differentially expressed genes (DEGs) were selected between the control and PP10-treated groups (Fig. 4A). Eighteen overlapping DEGs were identified between the two CRC cell lines, four of which are closely related to cancer progression, namely FGF1, THBS2, DAPK2, and GSTM2 (Fig. 4B). In addition, DEGs were analyzed using KEGG and GO analysis. GO enrichment analysis showed that PP10 impacts cell motility and extracellular matrix, processes related to migration and metastasis (Fig. 4C,D). KEGG analysis indicated the most enrichment pathway in PP10-treated cells was PI3k–AKT signaling pathway (Fig. 4E,F). Moreover, GSEA analysis indicated that PP10 was associated with the PI3K–AKT signaling pathway, epithelial

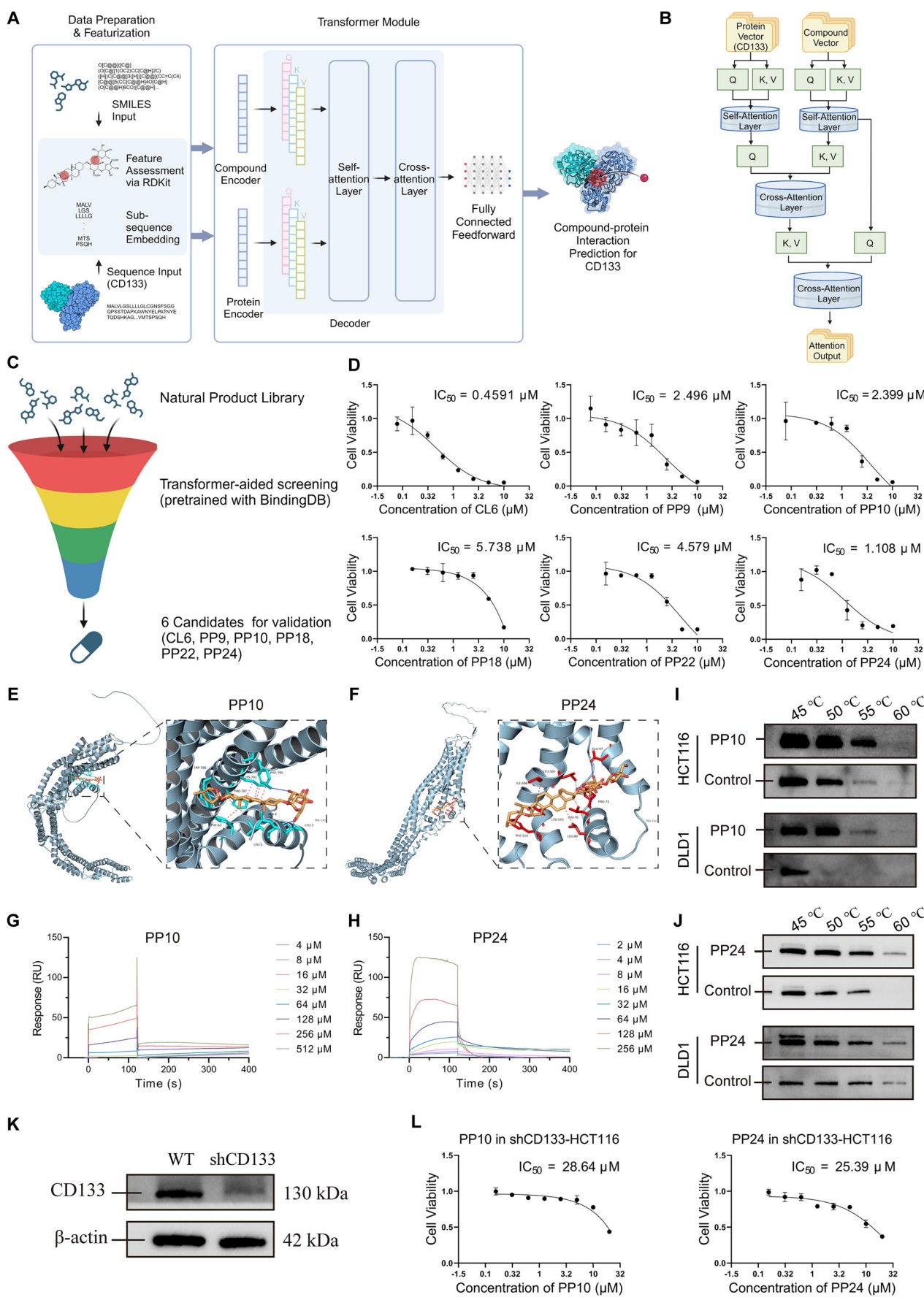

◄ **Figure 2.   Identification of CD133-targeted compounds via transformer-based CPI screening.**

(A) Workflow detailing the steps involved in data preparation, featurization, and the Transformer module application. (B) Detailed process of the Transformer-based algorithm for compound-protein interaction prediction. (C) Brief workflow of the compound screening process. (D) MTT assay results showing the cytotoxicity of CL6, PP9, PP10, PP18, PP22, and PP24 on HCT116 cells. (E, F) Molecular docking analysis of PP10, PP24 with CD133. Hydrophobic interactions are depicted in dotted lines, and hydrogen bonds are depicted in blue straight lines. (G, H) SPR results of the binding interactions between PP10, PP24, and CD133. (I, J) CETSA results showing the thermal stability of CD133 in the presence of PP10 and PP24. (K) Expression level of CD133 in wild-type and shCD133 HCT116 cell lines. (L) Cell viability assay results showing the cytotoxicity of PP10 and PP24 in shCD133-HCT116 cells. Student's $t$ test. Data are presented as mean ± SD. $^{*}P < 0.05$, $^{**}P < 0.01$, and $^{***}P < 0.001$ when compared with control group. Source data are available online for this figure.

cell-cell adhesion, and cellular oxidant detoxification (Fig. 4G). Compared with WT HCT116 cell, CD133-knockdown cell possessed weaker PI3K–AKT pathway activity, which indicated the potential to inhibit the PI3K–AKT pathway to prevent cancer cell growth (Fig. EV1B). To validate the transcriptome results, the protein expression levels of components in the PI3K–AKT pathway were examined (Ganesan et al, 2024). PP10 significantly reduced both the total and phosphorylation levels of PI3K, AKT, mTOR, and p65 (Figs. 4H,I and EV4A).

## PP10 promotes ROS overproduction to inhibit CRC cells' growth via inducing mitophagy blockage and pyroptosis

Based on GSEA results, PP10 was associated with elevated oxidative stress in CRC cells. ROS was assessed utilizing the DCFH-DA probe (Guo et al, 2022; Shi et al, 2024). As expected, PP10 promoted ROS overproduction, as evidenced by the increased green fluorescence, which was significantly scavenged by N-acetyl-l-cysteine (NAC) (Fig. 5A). Quantitative ROS detection using flow cytometry also corroborated these findings (Fig. 5B). Elevated ROS levels can cause mitochondrial damage and alter membrane permeability, which are key factors in inducing mitophagy (Hawk et al, 2018; Wang et al, 2023; Zhi et al, 2023). Mitophagy-related proteins were subsequently analyzed. PP10 suppressed Bcl-2 expression, while increasing Bax, P62, and stimulated the transformation of LC3B I to LC3B II. Chloroquine (CQ) treatment inhibited autophagic degradation, leading to the accumulation of LC3B and P62 in co-treated cells. Bafilomycin A1 (Baf-A1) was also used to inhibit autophagosome-lysosome fusion and cargo sequestration. The results showed a slight increase in LC3B II in PP10 and Baf-A1 group compared with the normal group. Based on the change of P62, it indicated the blockage of mitophagy in the terminal stage. Finally, the damaged mitochondria could not be removed and accumulated in cell resulting cell death. These results indicated PP10 could activate apoptosis and prevent mitophagy in the terminal stage to inhibit CRC cells growth (Figs. 5C,D and EV4B–D). To further investigate mitochondrial autolysosomes formation, a co-localization assay of mitochondria and lysosomes was conducted. Confocal microscopy demonstrated the integration of mitochondria (red) with lysosomes (green), resulting in the increasing of yellow spots in PP10-treated cells, indicating mitochondrion autolysosome formation (Figs. 5E and EV4E). LC3B fluorescence assays confirmed the transformation from LC3B I to LC3B II, implying autophagosome-lysosome integration. The strong red fluorescence intensity indicated substantial LC3B II recruitment to autophagosome membranes (Figs. 5F and EV4F). Transmission electron microscopy (TEM) further confirmed PP10-induced mitophagy (Fig. 5G). Control mitochondria exhibited well-

defined structures, whereas PP10-treated mitochondria showed autophagosomes with multilayer membranes, distorted cristae, and single-membrane autolysosomes, confirming mitophagy induction.

In addition, features of pyroptosis, including cell swelling and membrane vacuoles, were observed by TEM. The PP10-treated cells revealed membrane disarray and holes (Li et al, 2022) (Fig. 5H). Pyroptosis-related proteins analysis showed that PP10 enhanced cleaved Caspase-1 levels, which further cleaved GSDMD-F to form GSDMD-N. GSDMD-N is essential for forming membrane pores, leading to cellular content leakage and cell death. Cleaved Caspase-1 also processed pro-IL-1β into mature formation, thus promoting inflammation (Figs. 5I and EV4G,H). Disulfiram (DSF), an anti-pyroptosis agent, mitigated the pyroptotic effects induced by PP10. Collectively, these results confirmed that PP10 enhanced ROS overproduction to suppress CRC cells growth via inducing mitophagy and GSDMD-related pyroptosis.

## PP24 exerts anti-tumor effect by inhibiting the Wnt/β-catenin pathway

Transcriptome was first conducted to identify DEGs following PP24 treatment, and the results were analyzed using KEGG pathway enrichment. The Wnt/β-catenin signaling pathway was regarded as an important pathway influenced by PP24 treatment (Figs. 6A,B and EV5G). Compared with WT HCT116 cells, CD133-knockdown cells exhibited lower Wnt/β-catenin pathway activity (Fig. EV1C), providing a rationale for targeting this pathway in cancer therapy. Subsequently, the expression of key components was then assessed. PP24 treatment significantly decreased the cellular levels of phosphorylated AKT, GSK3β, Dvl2, Naked1, and β-catenin, and subsequently suppressed the nuclear levels of LEF1 and TCF1/TCF7. These changes inhibited the downstream transcription and expression of genes involved in proliferation, migration, and invasion. The addition of HLY78, a powerful activator of the Wnt pathway, rescued the PP24's inhibition effects, and restored the Wnt pathway (Figs. 6C and EV5A). ROS levels were also indicated a significant rise after PP24 treatment, which was reversed by adding NAC (Figs. 6D,E and EV5B,C). To further evaluate the apoptotic effects of PP24, an Annexin V/FITC staining assay was performed via flow cytometry. Treatment with PP24 at concentrations as low as 0.5 μM for 24 h induced apoptosis in HCT116 cells, while concentrations above 2 μM triggered significant apoptosis, with a higher percentage of late apoptotic cells. A similar trend was observed in DLD1 cells, indicating that PP24 induced apoptosis in a concentration-dependent manner (Figs. 6F,G and EV5D,E). To validate the apoptotic signaling pathway, the expression levels of key apoptosis markers were assessed. In PP24-treated HCT116 cells, there was a marked decline in Bcl-2, while a

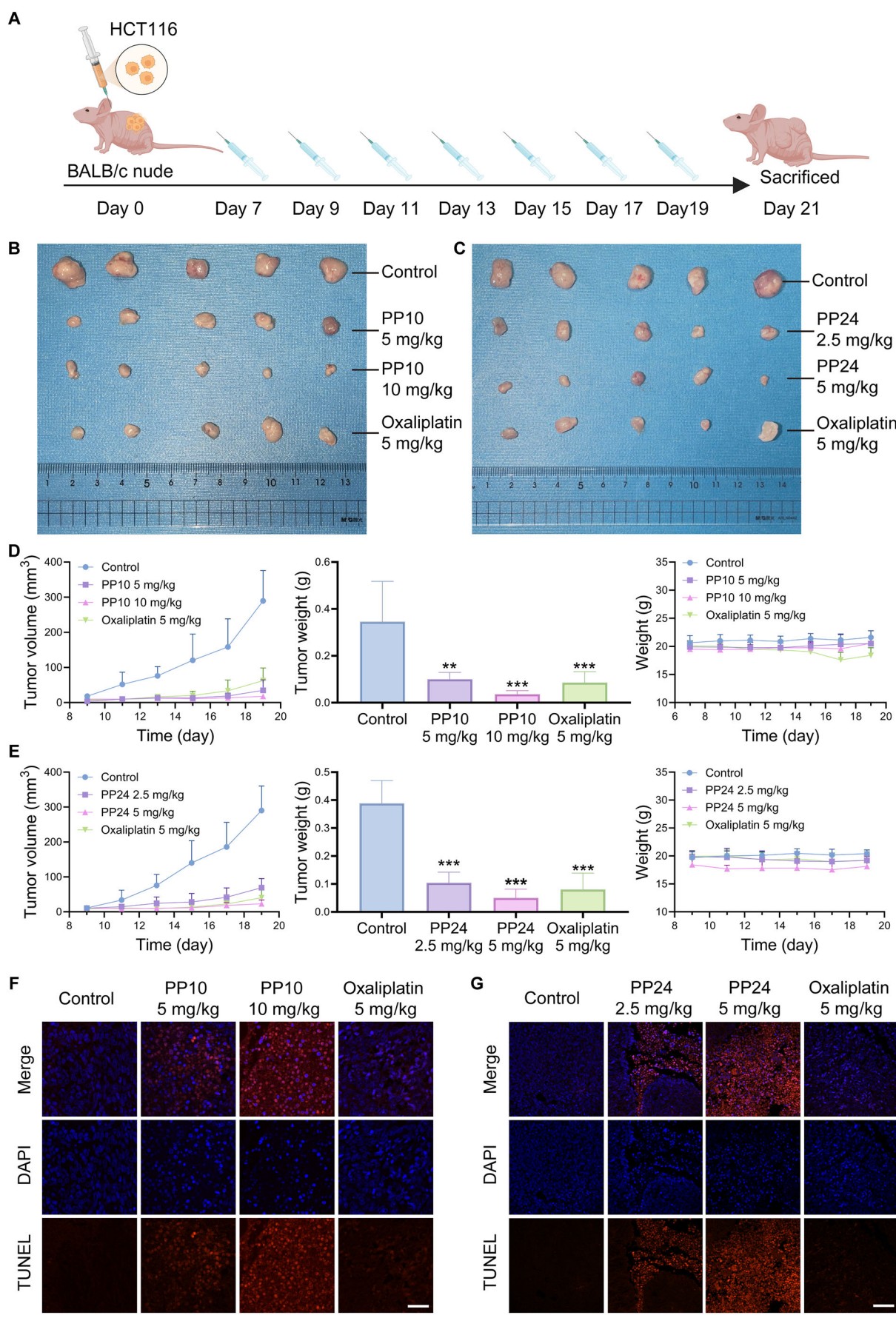

**Figure 3. PP10 and PP24 inhibited CRC growth in the CDX model with low toxicity.**

(A) Schematic workflow of model establishment and experiment design ($N = 5$). (B, C) Dissected tumor after PP10 and PP24 experiment. (D, E) Tumor volume, statistics of tumor weight, and body weight of PP10- and PP24-treated mice during the experiment ($N = 5$). (F, G) TUNEL staining after PP10 and PP24 treatment. Scale bar, 50 μm. one-way ANOVA test. Exact $P$ values are presented in Appendix Table S1. Data are presented as mean ± SD. $^*P < 0.05$, $^{**}P < 0.01$, and $^{***}P < 0.001$ when compared with control group. Source data are available online for this figure.

concomitant rise in Bax can be seen (Figs. 6H and EV5F). These results demonstrated that PP24 promoted apoptosis by blocking the Wnt/β-catenin signaling pathway.

## PP10 and PP24 inhibit the viability of pan-cancer organoids

In cancer research, various models are crucial for understanding disease mechanisms and evaluating therapeutic strategies. Traditional cell line models, while convenient, have limitations in replicating the complexity of tumors. Animal models provide valuable in vivo insights but often fail to fully replicate human pathophysiology. In contrast, patient-derived organoids (PDOs), with three-dimensional structures, offer a more accurate representation of tumor heterogeneity and microenvironmental dynamics (Tuveson and Clevers 2019; Chen et al, 2024). These PDO models provide an effective platform for assessing drug efficacy and toxicity with higher precision. Thus, we firstly constructed CRC, lung cancer, and liver cancer organoids using the droplet-engineered method to investigate the anticancer capability of PP10 and PP24. The results showed that PP10 and PP24 effectively inhibited organoids activity in all groups, with significant inhibition observed even at low concentrations (Fig. EV6A–C). Subsequently, different PDOs were sectioned and stained for CD133, AKT, and β-catenin to confirm the relevance of PP10 and PP24 treatment (Jiang et al, 2020). Immunofluorescence results revealed relatively high levels of CD133 expression in all the PDOs, supporting the proper use of PP10 and PP24 as CD133-targeted drugs for pan-cancer therapy. However, the different expression level of CD133 in different cancers of PDOs leaded to various therapeutical outcome of PP10 and PP24 in the following PDOX models. AKT and β-catenin were also over-expressed in pan-cancer organoids, suggesting that both pathways are crucial for cancer development (Fig. EV6D). To investigate the effects of PP10 and PP24 on signaling pathways in colorectal cancer PDOs, immunofluorescence staining was performed to assess proteins associated with cell proliferation and death. The results showed a significant reduction in Ki67 expression, indicating suppressed proliferative activity in the organoids. In addition, the decreased expression of AKT and β-catenin reflected the inhibition of the PI3K–AKT and Wnt/β-catenin signaling pathways, respectively. Furthermore, elevated levels of Caspase-1 in PP10-treated PDOs and Caspase-3 in PP24-treated PDOs suggested the induction of pyroptosis and apoptosis (Fig. EV7A,B). These results confirmed the anti-tumor effects of PP10 and PP24 in pan-cancer organoids via different cytotoxic mechanisms.

## PP10 and PP24 inhibit pan-cancer growth in a patient-derived organoid xenograft model

The pan-cancer PDOX models were constructed to investigate the anti-tumor activities for PP10 and PP24 in vivo by transplanting PDO into the subcutaneous region of NSG mice (Xiang et al, 2024).

The mice were separated into control, low-dose, and high-dose groups. In the CRC-PDOX experiment, compared to the control group, tumors in the treated groups showed size reduction, with a significant decrease in tumor weight observed in a dose-dependent manner. The PP24-treated group showed more pronounced suppression of tumor growth. Tumor growth curves revealed a remarkable suppression of PP10 and PP24 on tumor growth rates (Fig. 7A–E). To assess the influences of PP10 and PP24 on cellular pathways and cell death in CRC-PDOX, immunohistochemical staining for tumor tissue sections was conducted. A decrease in the expression of Ki67, β-catenin, and AKT was observed, indicating suppressed tumor cell proliferation. Elevated levels of Caspase-1 and Caspase-3 indicated the induction of pyroptosis and apoptosis of Polyphyllin groups (Appendix Fig. S4A,B). In addition, the liver cancer and lung cancer PDOX models also showed significant tumor growth suppression at the Polyphyllin groups, as PP24 demonstrating superior tumor-suppressive effects (Fig. 7F–K). PP24 effectively inhibited tumor growth in both CRC and liver cancer PDOX models, indicating potent anti-tumor activity. Finally, liver and kidney function tests revealed no significant changes in serum levels (Appendix Figs. S4C,D, 5B, and 6B), and tissue structure remained morphologically normal in the pan-cancer groups (Appendix Figs. S5A and 6A). Body weight measurements during the experiment also demonstrated the safety of PP10 and PP24 (Appendix Figs. S5C and 6C), with no abnormal reactions observed in the mice throughout the study. In conclusion, these animal studies confirmed the excellent biosafety profile and potent anti-tumor effects of PP10 and PP24.

## Discussion

In this study, the BindingDB database was used to train and develop a Transformer-based CPI screening model, which predicted candidate compounds from a natural product library targeting CD133. The model initially predicted 29 compounds targeting CD133 with comparable scores, while only 6 compounds exhibited strong cytotoxicity, indicating the limitations of virtual screening predictions. Challenges persist, including differences in bioactivity and the necessity for further mechanistic investigations. Subsequently, PP10 and PP24, extracted from *Paris polyphylla*, were selected for further investigation. PP10 and PP24 exhibited high binding affinity to CD133, with different binding sites leading to distinct anti-tumor mechanisms. Different binding sites lead to varying degrees and directions of conformational alterations, thereby affecting the interactions between the proteins and other molecules. For instance, after being activated by different small molecules, the conformational changes in the transmembrane domains of G protein-coupled receptors (GPCRs) determine their binding to G proteins or arrestins (Duan et al, 2023). The PP24 exhibited stronger tumor cytotoxicity, while PP10 owns more

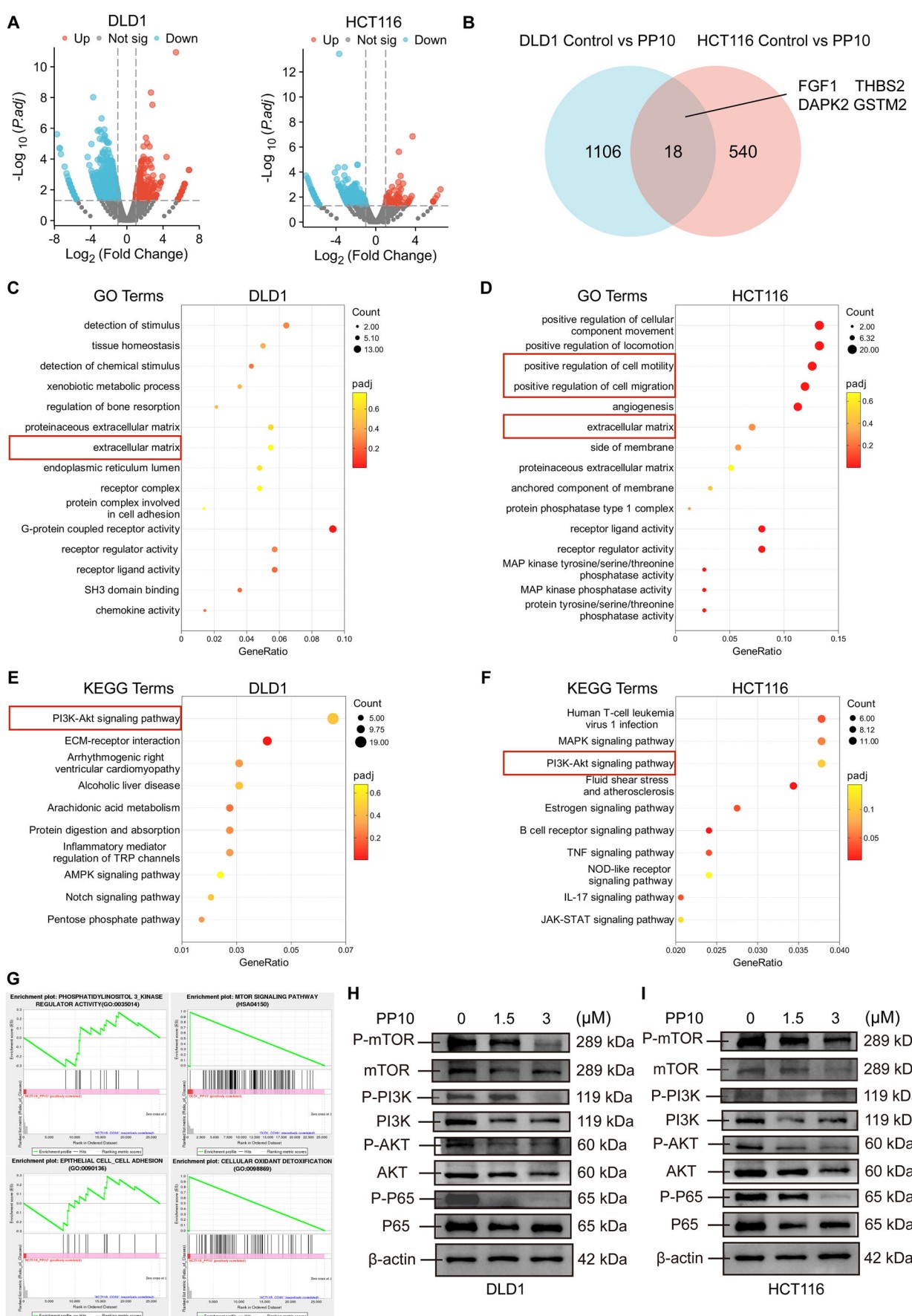

**Figure 4. Transcriptomic analysis of the impact of PP10 on CRC cells.**

(A) Volcano plot of DEGs associated with PP10 in HCT116 (upregulated DEGs = 89, downregulated DEGs = 481) and DLD1 cells (upregulated DEGs = 292, downregulated DEGs = 846) following PP10 treatment. (B) Overlapped DEGs associated with PP10 treatment in HCT116 and DLD1 cells. (C, D) KEGG pathway enrichment analysis of DEGs correlated with PP10 treatment in HCT116 and DLD1 cells. (E, F) GO function and pathway enrichment analysis of DEGs associated with PP10 treatment in HCT116 and DLD1 cells. (G) Gene Set Enrichment Analysis (GSEA) results showing pathway enrichment following PP10 treatment in HCT116 and DLD1 cells. (H, I) Western blot result of PI3K–AKT pathway and downstream transcription factors in the HCT116 and DLD1 cells. Student's $t$ test. Data are presented as mean ± SD. *$P < 0.05$, **$P < 0.01$, and ***$P < 0.001$ when compared with control group. Source data are available online for this figure.

potent inhibition in tumor cells migration and invasion. Animal experiments showed that PP10 and PP24 effectively inhibited tumor cells growth in vivo, with a more pronounced effect at higher doses when compared with clinical drug oxaliplatin. In the CRC PDOX model, extreme distinction was observed between two Polyphyllins.

Drug discovery in cancer therapeutics is intricate and time-consuming, encompassing the identification and validation of potential drug targets unique to the disease, followed by extensive testing of compounds for their efficacy and safety in variable models (Swanton et al, 2024). In this study, machine learning significantly reduced the time and cost of compound screening by identifying key CPI features or De novo drug design, enabling rapid identification of promising leads for subsequent biological validation (Doytchinova 2022). Notably, the different mechanisms have not yet been predicted by the virtual platform. This is because the AI-based model screens interactions based solely on the amino acid sequence or structures. It means that the model does not provide information about downstream pathway interventions following target inhibition. In this study, transcriptome sequencing and pathway enrichment analysis revealed that PP10 specifically impacts the PI3K–AKT signaling pathway, while PP24 affects the Wnt/β-catenin pathway. Although both of them are derived from *Paris polyphylla*, their analogous structures lead to distinct binding sites on CD133. This results in distinguished inhibition mechanisms targeting CD133. Moreover, PP10 primarily promotes CRC cells' death via blockage of mitophagy and pyroptosis, while PP24 predominantly induces apoptosis. In the future, machine learning could be used to study and train more cases, which would enhance the investigation of CPIs between different binding sites of target proteins and clarify downstream pathway alterations from a structural perspective. Thus, machine learning emerges as a valuable tool for advancing drug discovery and mechanistic research. However, it is imperative to emphasize that experimental validation remains a cornerstone, as it furnishes meaningful data that paves the way for deep learning, which complements the model.

CD133 is a promising pan-cancer target, with 19 drugs currently in preclinical or phase I trials, most of which are antibody-based or CAR-T cell therapies, such as CD44/CD133-targeted CAR-T and OXS-1650 (Waldron et al, 2011; Zhai et al, 2024). Cancer patients exhibit a high degree of heterogeneity, with differences in gene expression, metabolic characteristics, which poses a challenge for drug development targeting CD133. Moreover, although CD133 is a potential drug target in pan-cancer, designing drugs that accurately and effectively target CD133 remains a challenge. The drugs need to possess high specificity and affinity to ensure precise action on CD133-positive cells, while avoiding damage to normal cells. The aforementioned reasons are the primary obstacles hindering the development of CD133-targeted drugs. Natural products offer unique advantages in drug discovery, including

chemical diversity, flexible interactions with target proteins, and exceptional bioactivity (Atanasov et al, 2021). The current problem is that identifying natural product anti-tumor candidates through traditional methods is highly inefficient. Thus, we used novel methodologies, such as machine learning to expedite the screening process, and the results demonstrate that TransformerCPI can greatly improve the accuracy and efficiency of drug screening (Arora et al, 2024; Sim et al, 2025).

PP10 and PP24 have been studied for their potential applications in antiplatelet aggregation and the suppression of CYP450 activity (Li et al, 2024; Wang et al, 2024). However, their anti-tumor activity and targets across pan-cancer have not yet been investigated. This study is the first to conduct multi-platform validations of the roles of PP10 and PP24 in pan-cancer, potentially providing a broader theoretical foundation for the application of *Paris polyphyllin* in cancer therapy. Our results fully demonstrate that PP10 and PP24 can specifically bind to CD133 to achieve pan-cancer therapy. Specifically, PP10 can induce autophagy and pyroptosis via blocking the PI3K/AKT pathway, while PP24 mainly promotes cancer cell apoptosis via suppressing the Wnt/β-catenin pathway. This research may further elucidate their potential anticancer activities and strongly support the development of novel anticancer drugs.

This study aims to systematically and comprehensively investigate the anti-tumor activity, mechanisms, and biosafety of PP10 and PP24 using cancer cell lines, PDOs, and animal models. PDOs are derived from patient tumor tissues and cultivated in vitro to recreate the 3D tumor microenvironment, they have emerged as a transformative paradigm in personalized medicine, providing a versatile and reliable platform for advancing our understanding of cancer biology and therapeutic strategies (Jiang et al, 2020; Zhi et al, 2023; Ma et al, 2024). The PDOs possess certain tumor heterogeneity and complexity, and can retain the pathological characteristics, genetic profiles, mutation status, and drug responsiveness of the source tissue in terms of structure and function (Kuo and Curtis 2018; Tuveson and Clevers 2019; Herpers et al, 2022). Furthermore, PDOs have been shown to exhibit superiority over cell lines with respect to parental tumor representativeness and over xenografts with regard to cost-effectiveness and scalability (Vlachogiannis et al, 2018; Mo et al, 2022; Xu et al, 2022). PP10 and PP24 exhibit a promising tumor reductive effect on the PDO and PDOX model, providing a better understanding of their efficacy and biosafety profile for further clinical trial investigation. In particular, microfluidics-engineered cancer organoids, as demonstrated in this study and previous works, have proven their intrinsic high-throughput evaluation capacity and potential for automated operation(Jiang et al, 2020; Yang et al, 2024). The convergence of engineered organoid platforms with AI screening platforms is poised to transform the landscape of drug development.

## Limitations of the study

While this study demonstrates the potential of machine learning in accelerating CD133-targeted drug discovery, several limitations warrant consideration. First, the TransformerCPI model predicted 29 candidate compounds from a natural product library, yet only 6 exhibited strong cytotoxicity, highlighting inherent limitations of virtual screening in fully capturing bioactivity and mechanistic complexity. This discrepancy may stem from the model's reliance on amino acid sequences or structural data, which fail to predict downstream pathway interventions or binding site-specific effects. For instance, although PP10 and PP24 showed distinct mechanisms (PI3K–AKT vs. Wnt/β-catenin pathway inhibition), these differences were identified post hoc via transcriptomics rather than predicted by the AI platform. Current models lack mechanistic insights derived from structural interactions, necessitating laborious transcriptomic or proteomic validation to clarify functional impacts (Baselious et al, 2024; Cao et al, 2025). Second, CD133-targeted drug development faces challenges in addressing tumor heterogeneity and ensuring specificity. While PP10 and PP24 demonstrated pan-cancer efficacy in preclinical models, patient-specific genetic and metabolic variability may limit their broad applicability, necessitating personalized validation in diverse cohorts. Third, although PDO/PDOX models retained tumor heterogeneity and improved preclinical relevance compared to traditional cell lines, their predictive power for clinical outcomes remains unproven. Discrepancies between organoid responses and human trials, as noted in prior studies, underscore the need for further validation (Urrestizala-Arenaza et al, 2024). Future studies should integrate multi-omics data into AI models to predict mechanistic outcomes and prioritize compounds with balanced efficacy-specificity profiles. Additionally, expanding training datasets to include diverse binding sites and downstream pathway effects could enhance model robustness, bridging the gap between virtual screening and biological complexity.

## Methods

### Reagents and tools table

| Reagent/resource | Reference or source | Identifier or catalog number |
| --- | --- | --- |
| **Experimental models** | | |
| BALB/c Nude (*M. musculus*) | Guangdong Medical Laboratory Animal Center | NA |
| NSG Nude (*M. musculus*) | Guangdong Medical Laboratory Animal Center | NA |
| **Antibodies** | | |
| Anti-Akt | ABcolonal | Cat# A22770 |
| Anti-Bax | ABcolonal | Cat# A12009 |
| Anti-Bcl-2 | ABcolonal | Cat# A19693 |
| Anti-Caspase-1 | ABcolonal | Cat# A0964 |
| Anti-Caspase-3 | ABcolonal | Cat# A11319 |
| Anti-CD133 | ABcolonal | Cat# A28001 |
| Anti-Dvl2 | ABcolonal | Cat# A23686 |

| Reagent/resource | Reference or source | Identifier or catalog number |
| --- | --- | --- |
| Anti-E-cadherin | ABcolonal | Cat# A24874 |
| Goat anti-mouse IgG (H + L), HRP conjugate | Transgen | Cat# HS201-01 |
| Goat anti-rabbit IgG (H + L), HRP conjugate | Transgen | Cat# HS101-01 |
| Anti-GSDMD | ABcolonal | Cat# A20197 |
| Anti-GSK3β | ABcolonal | Cat# A2081 |
| Anti-IL-1β | ABcolonal | Cat# A16288 |
| Anti-Ki-67 | CST | Cat# 9129 |
| Anti-LC3B | ABcolonal | Cat# A11282 |
| Anti-LEF1 | ABcolonal | Cat# A23458 |
| Anti-mTOR | ABcolonal | Cat# A11355 |
| Anti-Naked-1 | ABcolonal | Cat# A17799 |
| Anti-N-cadherin | ABcolonal | Cat# A0433 |
| Anti-P62 | ABcolonal | Cat# A11483 |
| Anti-p-Akt | ABcolonal | Cat# AP1332 |
| Anti-p-GSK3β | ABcolonal | Cat# AP1341 |
| Anti-PI3K | ABcolonal | Cat# A4992 |
| Anti-p-mTOR | ABcolonal | Cat# AP0978 |
| Anti-p-PI3K | ABcolonal | Cat# AP0427 |
| Anti-Snail | ABcolonal | Cat# A11794 |
| Anti-TCF1/7 | ABcolonal | Cat# A20835 |
| Anti-Vimentin | ABcolonal | Cat# A19607 |
| Anti-β-actin | ABcolonal | Cat# AC038 |
| Anti-β-catenin | ABcolonal | Cat# A11512 |
| **Recombinant proteins** | | |
| CD133 recombinant protein | MCE | Cat# HY-P76196 |
| **Chemicals, enzymes, and other reagents** | | |
| DMEM | Gibco | Cat# 8120076 |
| Fetal bovine serum | Qing Mu | Cat# mu001SR |
| Trypsin | TransGen Biotech | Cat# FG301-11 |
| Penicillin–streptomycin | Beyotime | Cat# ST488 |
| MTT | Beyotime | Cat# ST316 |
| Crystal Violet | Beyotime | Cat# C0121 |
| DMSO | Solarbio | Cat# D8371 |
| Annexin V-FITC/PI Kit | TransGen Biotech | Cat# FA101-01 |
| Protein Ladder | Thermo Fisher | Cat# 26617 |
| ECL Substrate | 4 A BIOTECH | Cat# 4AW011-20 |
| Protease Inhibitor Cocktail | MCE | Cat# HY-K0010 |
| BCA Protein Assay Kit | Thermo Fisher | Cat# 23225 |
| RIPA Lysis Buffer | Beyotime | Cat# P0013B |
| SDS-PAGE Loading Buffer | Beyotime | Cat# P0015L |
| PAGE Gel Preparation Kit | Epizyme | Cat# PG110-114 |
| Ethanol | Shanghai Lingfeng | Cat# 8032-32-4 |
| Glycine | Vetec | Cat# V900144 |

| Reagent/resource | Reference or source | Identifier or catalog number |
|---|---|---|
| Trizma Base | Vetec | Cat# V900483 |
| PVDF | Bio-Rad | Cat# 1620177 |
| Triton X-100 | Sangon Biotech | Cat# A110694 |
| 4% PFA | Solarbio | Cat# P1110 |
| DAPI | Southern Biotech | Cat# 0100-20 |
| DAB | ZSGB-BIO | Cat# ZLI-9017 |
| HE Staining Kit | Beyotime | Cat# HZ7070 |
| Tween-20 | Sangon Biotech | Cat# A100777 |
| Antibody Dilution Buffer | Beyotime | Cat# P0023A |
| Stripping Buffer | Beyotime | Cat# P0025 |
| BSA | Beyotime | Cat# ST023 |
| Antifade Mounting Medium | Beyotime | Cat# P0126 |
| Xylenes | Biosharp | Cat# BS926 |
| CPBS | Solarbio | Cat# C1010 |
| Hematoxylin and Eosin Staining Kit | Beyotime | Cat# C0105S |
| Neutral Balsam Mounting Medium | Sangon Biotech | Cat# E675007 |
| Amine Coupling Kit | Cytiva | Cat# BR100050 |
| CM7 Sensor Chip | Cytiva | Cat# 29147020 |
| Sodium Acetate | Cytiva | Cat# BR100349 |
| Sodium Hydroxide | Cytiva | Cat# SH31088 |
| Serum Assay Kits | Jiancheng | Cat# A019-030 |
| TUNEL Apoptosis Assay Kit | Beyotime | Cat# C1086 |
| CellTiter-Glo 2.0 Cell Viability Assay Kit | Promega | Cat# G9241 |
| ROS Assay Kit | Beyotime | Cat# S0033 |
| **Software** | | |
| ImageJ | NIH | ImageJ 1.53k |
| SYBYL | Tripos | SYBYL 8.0 |
| PyMol | Schrodinger | PyMol 2.5.8 |
| GraphPad Prism | GraphPad Software, Inc. | GraphPad Prism 8.01 |
| Adobe Illustrator | Adobe Systems | Adobe Illustrator 2021 |
| FlowJo | Beckman | FlowJo 10 |
| **Other** | | |
| Cell Incubator | Thermo Fisher | Cat# 3100 |
| Biological Safety Cabinet | Esco Lifesciences | Cat# 2010655 |
| Refrigerated Centrifuge | Thermo Fisher | Cat# Micro 21 R |
| Ultracentrifuge | Beckman Coulter | Optima XE-100 |
| High Pressure Sterilizer | Tomy | Cat# SX-700 |
| Inverted Fluorescence Microscope | Nikon | Cat# TE2000 |
| Confocal Microscope | Olympus | Cat# FV1000 |
| Microplate Reader | BioTek | Synergy LX |

| Reagent/resource | Reference or source | Identifier or catalog number |
|---|---|---|
| Gel Imaging System | Bio-Rad | Cat# 1708370 |
| Metal Bath | DALB | Cat# GL-150B |
| Electrophoresis Apparatus | Bio-Rad | Cat# 1658033 |
| Electronic Scale | Sartorius | Cat# BSA2245 |
| Eppendorf Research Plus | Eppendorf | Cat# 3123000900 |
| Vernier Caliper | MNT | Cat# 0-150-2 |
| Ice Maker | Scientz Biotech | XB-150 |
| Syringe Pump | Lead Fluid | TYD01 |
| Flow Cytometer | Beckman Coulter | CytoFLEX |
| Cryostat Machine | Thermo Fisher | NX50 |
| Biacore T200 | GE | Cat# 28975001 |
| Cell Incubator | Thermo Fisher | Cat# 3100 |
| Biological Safety Cabinet | Esco Lifesciences | Cat# 2010655 |

## Cell lines culture

HCT116, DLD1, A549, HCCLM3, and CNE2, cell lines were obtained from the Cell Bank of the Shanghai Institutes for Biological Sciences, Chinese Academy of Sciences. These cell lines represent different cancer types: nasopharyngeal carcinoma (CNE2), colorectal carcinoma (HCT116 and DLD1), liver cancer (HCCLM3), and lung cancer (A549). DMEM medium (Gibco, 8120076) with 10% FBS (QmSuero, mu001SR) was utilized for cell culturing to provide essential nutrients for growth. The culture conditions were cultured at 37 °C and 5% $CO_2$ to mimic the in vivo environment. A 1% penicillin–streptomycin solution was added to prevent bacterial contamination. Typically, the cells reached 70–80% confluency within 24 h post-seeding, after which they were passaged or subjected to the required treatments.

## Animals

Female BALB/c mice (4–6 weeks old) and female NSG mice (4–6 weeks old) were acquired from the Guangdong Medical Laboratory Animal Center in China. These animals were maintained in the Laboratory Animal Center at Peking University Shenzhen Graduate School under specific pathogen-free conditions. Environmental conditions were controlled at $23 \pm 2$ °C, with humidity maintained between 50% and 65%, and a 12-h light/dark rhythm. The mice had unrestricted access to food and water. All animal procedures were authorized by the Animal Experimentation Ethics Committee of SIGS, Tsinghua University (Project 16, approved on 2021/11/10, protocol number 2021-KY-0875-003).

## Chemicals

The compounds used in this study are Polyphyllin V (PP10) and Polyphyllin H (PP24). Polyphyllin V (PP10, Henghuibio, HB21667-20 mg) has a molecular formula of $C_{39}H_{62}O_{12}$, a molecular weight of 722.90, and a CAS registration number of 19057-67-1. Polyphyllin H (PP24, MCE, HY-N2382) has a molecular formula

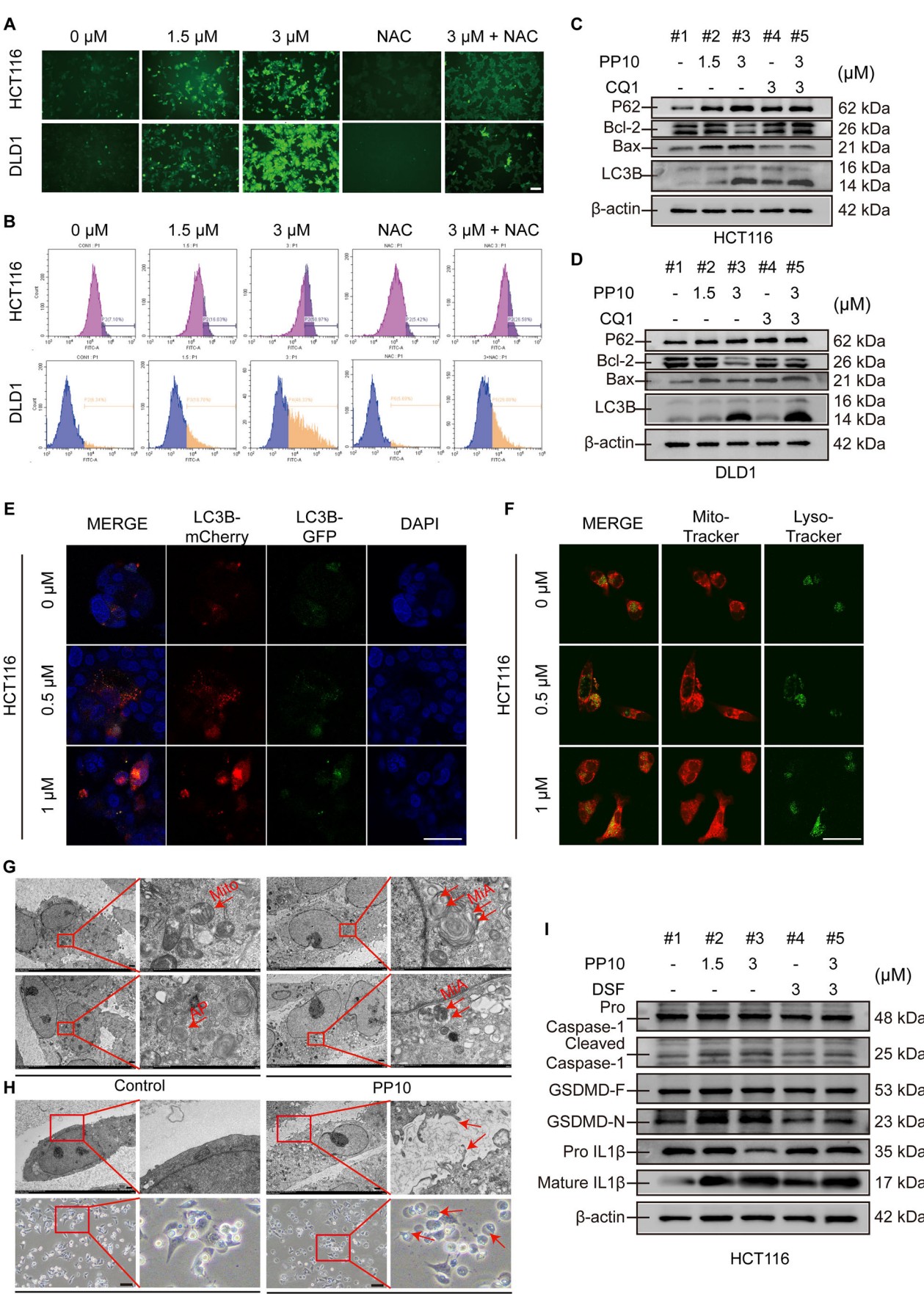

◄ **Figure 5.   PP10 promotes ROS overproduction to inhibit CRC cells growth via inducing mitophagy blockage and pyroptosis.**

(A) Microscopic observation of DCFH-DA probed cells treated with PP10, with or without NAC, showing ROS levels. Scale bar, 50 μm. (B) Flow cytometry analysis of ROS levels in CRC cells following PP10 treatment. (C, D) Western blot analysis of mitophagy-related proteins in HCT116 and DLD1 cells treated with PP10 and CQ1. (E) Microscopic observation of mCherry-GFP-LC3B infected HCT116 cells, showing autophagosome-lysosome fusion following PP10 treatment. Scale bar, 50 μm. (F) Microscopic observation of mitochondria and lysosome co-localization in HCT116 cells using Mito/Lyso-Tracker probes after PP10 treatment. Scale bar, 50 μm. (G) Transmission electron micrographs (TEM) showing mitochondrial changes in CRC cells after PP10 treatment. Mito mitochondrion, AP autophagosome, MiA mitochondrion autolysosome. (H) Microscopic and transmission electron micrographs showing pyroptosis features in CRC cells after PP10 treatment. Scale bar, 1 μm. (I) Western blot analysis of pyroptosis-related proteins in HCT116 cells treated with PP10. Student's $t$ test. Data are presented as mean ± SD. $^*P < 0.05$, $^{**}P < 0.01$, and $^{***}P < 0.001$ when compared with control group. Source data are available online for this figure.

of $C_{44}H_{70}O_{17}$, a molecular weight of 871.00, and a CAS registration number of 81917-50-2. Both compounds were white powder and dissolved in DMSO to prepare 10 mM stock solutions, aliquoted in 20 μL portions per tube, and stored at −80 °C. The stock solutions were dissolved in different solvents depending on the experimental requirements.

## Cell viability assay

The MTT assays were utilized to evaluate tumor cell viability by detecting the activity of succinate dehydrogenase in mitochondria. For the assay, $5 \times 10^3$ cells per well were cultured in 96-well plates in a cell culture incubator. After adhesion, the culture medium was refreshed with new medium including varying concentrations of the drugs (10, 5, 2.5, 1.25, 0.63, 0.315, 0.158 μM) for an additional 24 h of incubation. Next, 5 mg/mL of MTT (Aladdin, M158055) was introduced and incubated for 4 h to form formazan. After that, the supernatant was carefully discarded, and 150 μL of pure DMSO was used to solubilize the precipitate and evaluated at 490 nm utilized an EnSight Multimode Plate Reader (PerkinElmer, Singapore).

## Transcriptome sequencing

HCT116 and DLD1 cells were treated with 0, 3 μM PP10 or PP24 for 24 h. Total RNA was extracted with TRIzol, and mRNA was purified using poly-T oligo-conjugated magnetic beads. After cDNA synthesis and PCR amplification, libraries were constructed and sequenced on an Illumina NovaSeq 6000. Clean reads were obtained following quality control, aligned to the reference genome with HISAT2 (v2.0.5), and quantified by FeatureCounts (v1.5.0-p3). Gene-level FPKM values were calculated from gene length and read counts. Differential expression based on FPKM values between PP10 or PP24 and control groups was analyzed using DESeq2 (v1.20.0), and GO, KEGG, and GSEA enrichment analyses of DEGs were performed with clusterProfiler (v3.8.1).

## Bioinformatics analysis

The mRNA transcriptome data for PROM1 were obtained from the TCGA and GEO databases (GSE21815). Data visualization was performed using ggplot2 (version 3.3.3) and presented as a box plot. Survival data from CRC patients in the TCGA and GSE21815 datasets were then used to generate Kaplan–Meier curves reflecting overall survival rates. Survival analysis based on high and low expression of PROM1 was performed using RNA-seq data (version 3.2-10), with visualization carried out using the survminer package (version 0.4.9).

## TransformerCPI-based drug screening

In order to screen for potential antagonist of CD133, the Transformer-based algorithm was employed to predict the binding potential between CD133 and molecules from the natural product library. The model processes CD133 protein sequences by partitioning them into words and then converting them into real-valued embeddings using pre-trained word2vec models. For compounds, atom features are transformed into vectors using RDKit, followed by Graph Convolutional Networks (GCN) for learning representations. In TransformerCPI, protein and atomic representations are transformed for input into the model. The encoder replaces traditional self-attention layers with a one-dimensional convolutional gated linear unit (GLU) to mitigate overfitting. Multi-headed self-attention in the decoder including self-attention layers and feed-forward layers extracts interaction information, enhancing the model's performance in CPI prediction tasks. After comparison and ranking, a total of 29 compounds that have the potential to interact with CD133 were discovered.

## Molecular docking

The crystallographic structure of human CD133 (Isoform-I) was constructed by using AlphaFold, as no experimental structures are available in the PDB. Molecular modeling was performed using SYBYL 8.0 software (Tripos, USA), which generated the three-dimensional structures of PP10 and PP24 following energy minimization. During protein preparation, co-crystallized ligands and water molecules were excluded, and hydrogen atoms were added. The docking of PP10 and PP24 with the prepared protein structure was carried out using Surflex-Dock in SYBYL 8.0. The resulting ligand-receptor complex was visualized using PyMol 2.5.8 software (Schrodinger, USA).

## Apoptosis assay

In total, $2 \times 10^5$ HCT116 and DLD1 cells were seeded into six-well plates for 24 h. Then the cells were treated by varying concentrations of PP10 or PP24 (0, 0.8, 1.5, 3, 6 μM) for an additional 24 h of treatment. Next, the cells were washed with cold PBS. Cells were then digested with 0.25% trypsin without EDTA (TRANS, FG301-11), followed by washing with PBS. For apoptosis analysis, the Annexin V-FITC/PI Kit (4 A Biotech, FXP018) was utilized for staining according to the manufacturer's protocol. Apoptosis was measured by flow cytometry and analyzed with FlowJo (Beckman, USA).

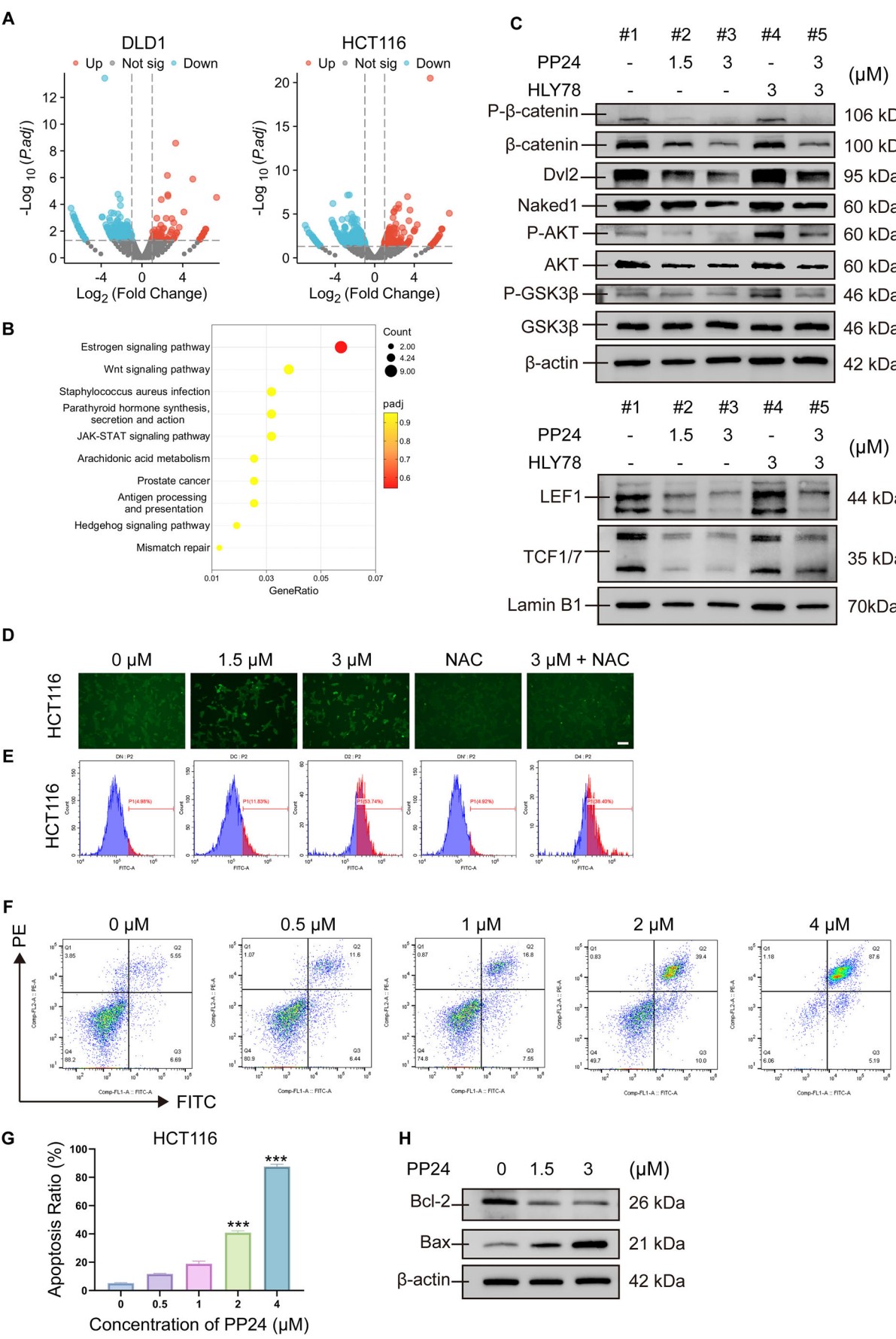

**Figure 6.  PP24 exerts anti-tumor effect via inhibiting the Wnt/β-catenin pathway.**

(A) Volcano plot of DEGs associated with PP24 in HCT116 (upregulated DEGs = 177, downregulated DEGs = 399) and DLD1 cells (upregulated DEGs = 238, downregulated DEGs = 588). (B) KEGG pathway enrichment analysis showing the functional pathways affected by DEGs in HCT116 cells following PP24 treatment. (C) Western blot analysis of Wnt/β-catenin pathway proteins and downstream transcription factors following PP24 treatment in HCT116 cells. (D) Microscopic observation of ROS levels in HCT116 treated with PP24 and/or NAC using DCFH-DA probe. (E) Flow cytometry results showing ROS detection after treatment with PP24 and NAC in HCT116 cells. (F, G) Flow cytometry analysis of apoptosis induction in HCT116 cells after treatment with PP24 (N = 3). (H) Western blot analysis of apoptosis-related pathway proteins in HCT116 cells following PP24 treatment. Scale bar, 50 μm. one-way ANOVA test. Exact P values are presented in Appendix Table S1. Data are presented as mean ± SD. $^{*}P < 0.05$, $^{**}P < 0.01$, and $^{***}P < 0.001$ when compared with control group. Source data are available online for this figure.

## Surface plasmon resonance (SPR)

CD133 protein was covalently immobilized on a CM7 chip (Cytiva, USA) using amino coupling. A concentration gradient of PP10 or PP24 was then injected and flowed over the chip surface. Binding of the molecules to CD133 induced a surface plasmon resonance (SPR) phenomenon, which was converted into an electrical signal, referred to as the response unit (RU). The equilibrium dissociation constant (KD) for the interaction between PP10 or PP24 and CD133 was determined using a Biacore T200 system (GE Healthcare, USA). Data were analyzed with Biacore Evaluation Software, which applies affinity and kinetic models to fit the data based on multiple concentration curves and calculates the corresponding KD values.

## Cellular thermal shift assay

CETSA was employed for the evaluation of the interaction between PP10 or PP24 and CD133. HCT116 and DLD1 cells were harvested, and liquid nitrogen was utilized for cell lysis. The resulting cell lysate was divided into two equal portions. One portion was treated with PP10 or PP24, while the other portion received DMSO as a control. After 1 h incubation at RT, the samples were exposed to different temperatures (45 °C, 50 °C, 55 °C, 60 °C) for 3 min and chilled by ice for 5 min. Following centrifugation, the supernatants were collected and detected by Western blotting for CD133 expression.

## Scratch wound healing assay

In all, $1 \times 10^5$ HCT116 and DLD1 cells were incubated in 12-well plates. As cell confluence reached 90–100%, a scratch was made using a 200 μL pipette tip. Floating cells were discarded. Fresh medium containing PP10 or PP24 (0, 0.25, 0.5, 1 μM) was then introduced. The cell migration distance was recorded at 0, 12, 24, and 48 h using a microscope (Nikon, Japan), and the images were analyzed using ImageJ.

## Colony formation assay

CRC cells were seeded into 6-well plates at a density of 300 cells per well. When colonies reached 16 cells in number, the medium was replaced with fresh medium containing PP10 or PP24 (0, 0.25, 0.5, 1 μM). After 2 weeks of culture, cells were fixed with 4% paraformaldehyde (PFA, Servivebio, G1101) and stained with 0.5% crystal violet (Sigma-Aldrich, C0775). Colony numbers were then counted.

## Transwell migration and invasion assay

For migration assays, $8 \times 10^4$ cells and varying concentrations of PP10 or PP24 (0, 0.5, 1 μM) were introduced to the upper chamber of the transwell, while medium containing 20% FBS was placed in the lower chamber to attract the cells. For invasion assays, Matrigel (Corning, USA) was pre-coated onto the upper chamber to mimic the extracellular matrix, creating conditions similar to the migration assay for measuring metastatic potential. After 24 h of incubation, transwell chambers were washed with PBS and fixed with 4% paraformaldehyde (PFA) for 20 min. Untransferred cells on the upper side of the membrane were removed using cotton swabs. The transferred cells on the lower side were stained with 0.5% crystal violet, and unbound dye was washed away. Finally, images were captured by a microscope, and the number of cells was counted.

## Western blot

Overall, $3 \times 10^5$ HCT116 and DLD1 cells were placed into 6-well plates for 24 h to allow adherence. The cells were treated with 0, 1.5, or 3 μM PP10 or PP24 for 24 h. After treatment, Protein was extracted by RIPA Lysis Buffer (MedChemExpress, HY-K1001) and concentration was determined by a BCA kit (Thermo Scientific, A53225). Separation of proteins was conducted by electrophoresis on a 12.5% SDS-PAGE gel (Epizyme, PG213). The proteins were then moved to PVDF membranes (BIO-RAD, 1620177) and incubated with 5% skim milk in PBS for 1 h at RT. The membranes were sectioned and incubated overnight with the primary antibody solution at 4 °C. Next, the membranes were introduced by species-specific HRP-conjugated secondary antibodies (1:4000) for 1 h at room temperature. Protein bands were visualized using the ECL system (4A BIOTECH, 4AW011-200) and imaged with a ChemiDoc XRS$^{+}$ Imaging System (BIO-RAD, USA).

The following primary antibodies (1:1000) were used: Caspase-1 (A0964), GSDMD (A20197), IL-1β (A16288), Caspase-3 (A19654), Bax (A12009), Bcl-2 (A19693), β-actin (AC038), p-mTOR (AP0978), mTOR (A11355), p-PI3K (AP0427), PI3K (A4992), p-Akt (AP1332), Akt (A22770), P62 (A11483), LC3B (A11282), p-P65 (AP1294), P65 (A22331), β-catenin (A11512), Dvl2 (A23686), Naked-1 (A17799), p-GSK3β (AP1341), GSK3β (A2081), LEF1 (A23458), TCF1/7 (A20835), CD133 (12711), N-cadherin (A0433), E-cadherin (A24874), Vimentin (A19607), and Snail (A11794), which were bought from ABclonal Technology Co. Ltd.,

## ROS assay

The level of ROS generated by cells was evaluated using DCFH-DA (Beyotime, S0033S). Overall, $2 \times 10^5$ CRC cells were placed in 6-well plates for 24 h to allow adherence. The cells were then treated with varying concentrations of PP10 or PP24 (0, 1.5, 3 μM) and NAC (5 μM). After treatment, the cells were incubated with 10 μM

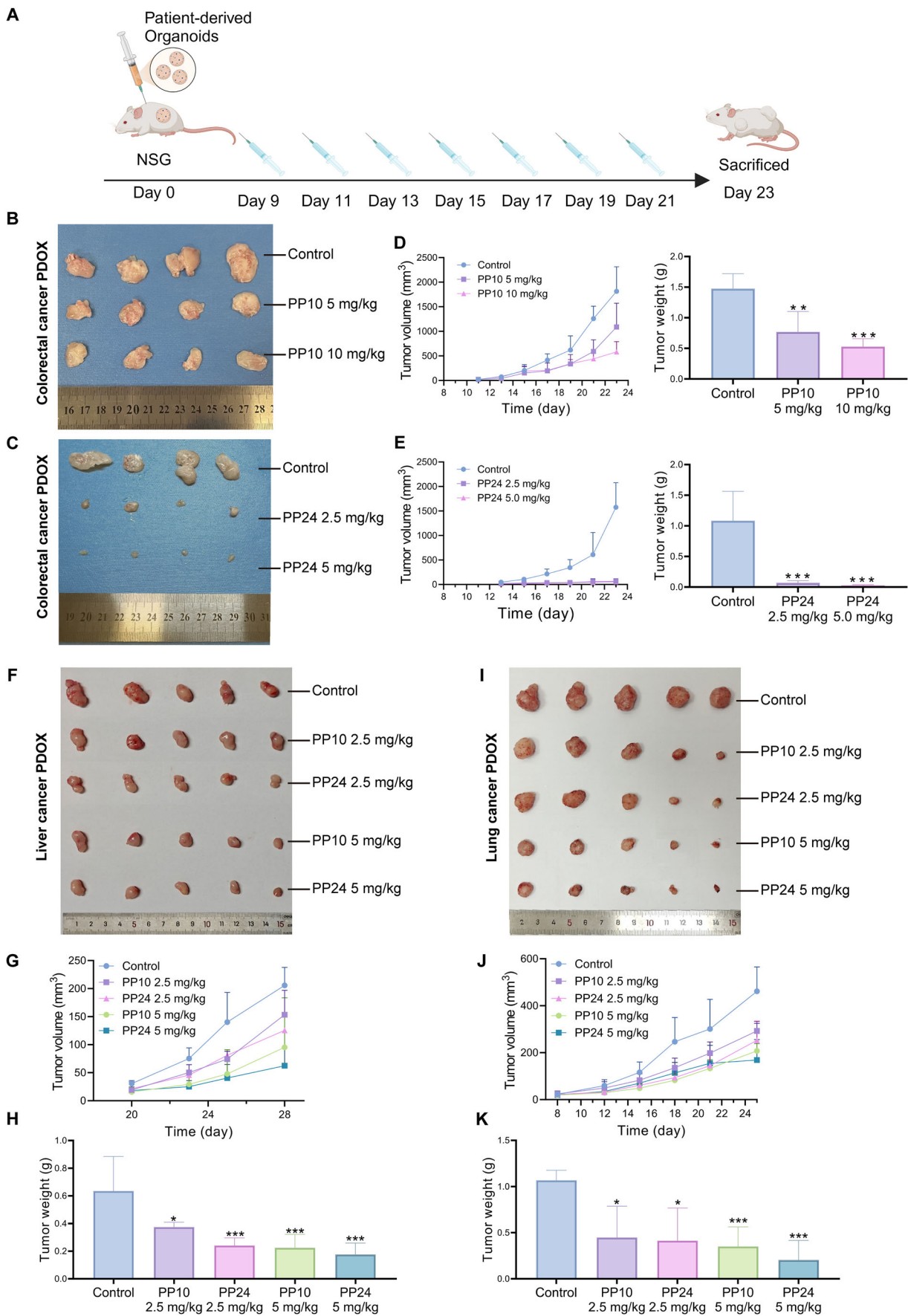

**Figure 7. PP10 and PP24 inhibited tumor growth in the pan-cancer PDOX model.**

(A) Schematic workflow of model establishment and experiment design ($N = 4$). (B, C) Dissected CRC tumor after PP10 and PP24 experiment. (D, E) Tumor volume and statistics of tumor weight of PP10 and PP24-treated mice during the experiment ($N = 4$). (F) Dissected tumor of liver cancer after PP10 and PP24 experiment. (G) Tumor volume of PP10 and PP24-treated liver cancer mice during the experiment ($N = 5$). (H) Statistics of tumor weight of PP10 and PP24-treated liver cancer mice during the experiment ($N = 5$). (I) Dissected tumor of lung cancer after PP10 and PP24 experiments. (J) Tumor volume of PP10 and PP24-treated lung cancer mice during the experiment ($N = 5$). (K) Statistics of tumor weight of PP10 and PP24-treated lung cancer mice during the experiment ($N = 5$). one-way ANOVA test. Exact $P$ values are presented in Appendix Table S1. Data are presented as mean ± SD. $^*P < 0.05$, $^{**}P < 0.01$, and $^{***}P < 0.001$ when compared with control group. Source data are available online for this figure.

DCFH-DA for 30 min at 37 °C. ROS levels were visualized by a fluorescence microscope and quantified using a flow cytometer.

## LC3B transfection and autophagy flux detection

LC3B is a key protein involved in cellular autophagy, and its fluorescence localization can be used to assess the stages and status of autophagy. In this experiment, mCherry-GFP-LC3B adenovirus (Beyotime, C3012-1 mL) was used to visualize autophagy. The mCherry-GFP-LC3B protein was transfected into the cells, and when autophagosomes fused with lysosomes, a change of pH causes a significant quenching of green fluorescence, while mCherry fluorescence remains unaffected. This fusion protein allows for tracking the activation of autophagy. CRC cells ($2 \times 10^4$) were seeded onto confocal dishes (Biosharp, BS-15-GJM) and cultured for 24 h. The cells were then infected with 20 multiplicities of infection (MOI) of the mCherry-GFP-LC3B adenovirus in 1 mL of fresh medium. After 12 h, the adenovirus-containing medium was discarded and replaced with fresh medium. After an additional 12-hour incubation, CRC cells were treated with 0, 0.5, or 1 µM PP10 for 24 h. The cells were then fixed with 4% paraformaldehyde (PFA), and an anti-fluorescence quenching mounting solution (Beyotime, P0126-5 mL) was applied. The autophagy status of CRC cells was observed by a fluorescence microscope (Nikon, Japan).

## Co-localization of mitochondria and lysosomes

Lysosomal and mitochondrial probes were used to visualize the spatial relationship between these two organelles after drug treatment to determine whether autophagosomes containing mitochondria fuse with lysosomes, thus confirming whether autophagic flux is unimpeded. CRC cells ($2 \times 10^4$) were seeded onto confocal dishes and treated with PP10 (0, 0.5, 1 µM). Next, cells were stained by Mito-Tracker and Lyso-Tracker (Beyotime, C1049B, C1047S) for 30 min. Unbound tracers were removed with PBS, and fresh medium was added to maintain the cell environment. Fluorescent images were immediately captured using a fluorescence microscope.

## Establishment of subcutaneous xenograft model in BALB/c nude mice

A cell-derived xenograft (CDX) model using BALB/c nude mice was employed to assess the efficacy and safety of PP10 and PP24 in vivo. $2 \times 10^6$ HCT116 cells were subcutaneously injected into the right hind limb for each mouse. Tumor growth was monitored every two days thereafter. On day 5 post-injection, mice were separated at random in control, 5 mg/kg PP10 or 2.5 mg/kg PP24,

10 mg/kg PP10 or 5 mg/kg PP24, and 5 mg/kg oxaliplatin ($n = 5$ per group). An equivalent volume of saline was administered to the control group. From day 7 post-injection, the mice were treated with the respective compounds (PP10, PP24, or oxaliplatin) via intraperitoneal injection bi-daily. Tumor volume was recorded using a vernier caliper, and body weight was recorded using balance bi-daily. On day 21, the experiment was terminated. Mice were anesthetized with $CO_2$, and blood samples were collected. Euthanasia was performed, followed by dissection of tumors and major organs (heart, liver, spleen, lung, and kidney). Tumor weight and organ conditions were recorded.

## Establishment of patient-derived tumor organoid xenograft model in NSG mice

NSG mice, known for their higher compatibility with human tumor tissues, were utilized in this study to create a patient-derived organoid xenograft (PDOX) model. Patient-derived tumor samples were obtained from Shenzhen People's Hospital, Shenzhen, China (Ethical Development No. 2024-08401). Patient tumor tissue was minced, digested, and mixed with matrix gel. Organoid microspheres of uniform size were then generated using a microfluidic system based on gas-liquid jetting. These organoids were subcutaneously implanted into the right abdomen of the mice, with monitoring conducted every two days. On day 9 post-implantation, mice were randomly assigned to three groups ($n = 4$ per group). For the colorectal cancer (CRC) PDOX model, the groups were: control, 5 mg/kg PP10 or 2.5 mg/kg PP24, and 10 mg/kg PP10 or 5 mg/kg PP24. Tumor volume and body weight were recorded bi-daily, and compounds were administered intraperitoneally. On day 23 post-implantation, the experiment was terminated. Mice were anesthetized with CO2, blood was collected, and euthanasia was performed. Tumors and major organs (heart, liver, spleen, lung, kidney) were dissected and weighed (Guo et al, 2016). For the liver cancer and lung cancer PDOX models, the groups were: control, 2.5 mg/kg PP10, 2.5 mg/kg PP24, 5 mg/kg PP10, and 5 mg/kg PP24.

## Evaluation of liver and kidney functions of animal models

To assess the drug's impact on the normal physiological functions of mice, liver and kidney function markers are measured in the blood. Blood samples are collected and left overnight at 4 °C. After centrifugation (4000 rpm for 30 min), the supernatant is stored at −20 °C until analysis. Liver function markers (AKP, ALT, AST) and kidney function markers (BUN, CREA, LDH) are measured using appropriate assay kits (Jiancheng, A059-2-2, C099-2-1, C010-2-1, C013-2-1, C011-2-1, A020-2-2) (Ren et al, 2021). Absorbance is detected with a microplate reader, and data are analyzed using GraphPad software.

## HE and IHC staining

Tumor and organ tissues are fixed in 4% paraformaldehyde (PFA) at room temperature for 48 h with gentle agitation. After fixation, the tissues are paraffin-embedded, sectioned, and stained with hematoxylin and eosin (HE) (Servicebio, China). For immunohistochemistry, tumor tissues are subjected to deparaffinization, rehydration, antigen retrieval, and blocking before staining with antibodies against Ki67, Caspase-1, Caspase-3, β-catenin, and AKT. Following incubation with a secondary antibody, 3,3'-diaminobenzidine tetrahydrochloride (DAB, Solarbio, DA1010) is used as a chromogen to visualize the expression of the target proteins.

## TUNEL assay

The situation of apoptosis of the tumor was evaluated by TUNEL Apoptosis Assay Kit (Beyotime, China). The tumor sections were deparaffinized, rehydrated, and fixed in 4% paraformaldehyde for 30 min. After fixation, the tissues were incubated with proteinase K for 30 min. A mixture of TdT enzyme and fluorescein-labeled dUTP was then prepared and applied to the tissue for 1 h of incubation at 37 °C. The nuclei were stained with DAPI (Sigma-Aldrich, D9542) for 10 min. Apoptotic cells were detected by a microscope.

## Organoid fabrication and culture

Organoids were generated from patient-derived tissues to assess the sensitivity of PP10 and PP24 in a model that more closely mimics the pathological environment. After collection, patient-derived tumor tissue was cut and digested to form cell suspensions. In total, 1500 tumor cells were then added to Matrigel per droplet. This mixture was sheared into monodisperse microdroplets using the hydrophobic phase of an electronic fluorinated solution, resulting in organoids with an average diameter of 500 μm. The organoid droplets were incubated at 4 °C for 15 min for solidification. Finally, the organoids were cultured in ultra-low attachment 6-cm dishes in an incubator.

## Organoid viability assay

The CellTiter-Glo 2.0 Assay Kit (Promega, USA) was utilized in this study to detect ATP levels and assess the viability of organoids following treatment with PP24. Organoids were transferred to ultra-low attachment 96-well plates and treated with various concentrations of PP24 (0, 0.75, 1.5, 3 μM) in culture medium for 7 days. Images of the organoids were captured using a microscope. After the treatment period, the 96-well plates were placed at RT for 30 min to equilibrate, as the CellTiter reagent was preheated to 22 °C. An equal volume of the reagent was then introduced to the organoids, followed by shaking for 2 min. The plates were incubated at RT for 10 min. Fluorescence intensity was recorded by a microplate reader.

## Frozen section

To preserve the morphology of the small organoids, OCT (Optimal Cutting Temperature) compound was used for embedding, followed by sectioning with a cryostat. The organoids were placed into embedding molds, and excess liquid was removed before incubating them in a 20% sucrose solution for 2 h. After removing

the excess sucrose, an appropriate amount of OCT was injected, and the position of the organoids was adjusted accordingly. The embedding molds were then placed in a −80 °C freezer overnight. After freezing, the OCT-embedded organoid blocks were removed and placed in the cryostat. Once the samples were securely fixed, they were sectioned into slices of 8–10 μm thickness. The frozen sections were left at room temperature for 1 h to allow fixation, then stored at −20 °C for preservation.

## Immunofluorescence staining of organoids

Initially, the frozen sections were placed at RT for 1 h to prevent detachment. The samples were rinsed by PBS for 10 min to rehydrate, followed by PBS washing. Next, the sections were incubated with a blocking solution for 1 h. After discarding the blocking solution, the sections were incubated with a suitable amount of primary antibody solution overnight at 4 °C. After incubation, the sections were brought to RT for 30 min and washed three times with PBS. Then, the sections were incubated with the fluorescently labeled secondary antibody corresponding to the primary antibody species for 1 h. After washing three times with PBS, the nuclei were stained with DAPI for 5 min. Finally, the results were captured using a fluorescence microscope.

### The paper explained

**Problem**

Pan-cancer therapies targeting CD133 are limited by the complex structure of CD133 and tumor heterogeneity. Current therapeutic approaches face challenges such as drug shortages, off-target effects, toxicity, and drug resistance. There is an urgent need for efficient screening methods to identify CD133-targeted drugs. Virtual screening, despite its limitations in predicting downstream pathway interventions, is crucial for efficiently identifying potential CD133-targeted drugs and accelerating therapeutic discovery.

**Results**

The authors conducted bioinformatics analyses revealing elevated CD133 expression in several cancers, including CRC, and its link to poor prognosis. Using TransformerCPI, we screened 1123 compounds and identified Polyphyllin V and H as CD133-targeting candidates. These compounds demonstrated significant cytotoxicity in CRC cell lines. Molecular docking and SPR experiments confirmed their binding affinity to CD133. Moreover, the compounds inhibited CRC cell migration and invasion in vitro, and remarkably reduced tumor volume and weight in cell-derived xenograft models with low toxicity. Mechanistic studies showed that Polyphyllin V suppresses the PI3K–AKT pathway, inducing mitophagy blockage and pyroptosis, while Polyphyllin H inhibits the Wnt/β-catenin pathway, triggering apoptosis. In addition, patient-derived organoid xenograft models confirmed the broad anti-tumor effects of Polyphyllin V and H across multiple cancer types, including colorectal cancer, liver cancer, and lung cancer.

**Impact**

This study demonstrates the potential of AI-driven screening combined with biological validation for identifying novel CD133-targeted drugs. The identification of Polyphyllin V and H as effective anti-tumor agents with distinct mechanisms provides a promising foundation for pan-cancer therapy. The research also underscores the limitations of virtual screening alone and emphasizes the necessity of live model evaluation in AI-based therapeutic discovery.

## Statistical analysis

Statistical analysis in this study was conducted using GraphPad Prism 10.0.0 software. A Student's *t* test was applied to analyze comparisons between two groups, while one-way ANOVA was used for analysis of multiple groups. Multiple comparisons were performed to compare each experimental group with the control group. The results are presented as mean ± SD. Exact *P* value are presented in Appendix Tables S1–3. Statistical significance is indicated as follows: *$P < 0.05$, **$P < 0.01$, and ***$P < 0.001$.

## Graphics

Synopsis graphics and some of the figures were created with BioRender.com.

# Data availability

The datasets produced in this study are available in the following databases: RNA sequence Raw data: Gene Expression Omnibus GSE300390.

The source data of this paper are collected in the following database record: biostudies:S-SCDT-10_1038-S44321-025-00308-1.

# Peer review information

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

## Acknowledgements

The work was funded by the National Key-Area Research and Development Program of China (2024YFA0919800); National Natural Science Foundation of China (32371470 and 82341019); Shenzhen Fundamental Research Program (No. JCYJ20240813112004006); Shenzhen Major Science and Technology Projects (KJZD20230923115400001); Shenzhen Fundamental Research Program (No. JCYJ20240813112004006); Merck Research Grant, and the Cross-disciplinary Research and Innovation Fund of Tsinghua SIGS (No. JC2022007); Guangdong Basic and Applied Basic Research Foundation (2023B1515120025).

## Author contributions

**Yibo Hou**: Data curation; Formal analysis; Validation; Investigation; Visualization; Writing—original draft. **Zixian Wang**: Formal analysis; Validation; Investigation; Writing—original draft. **Wenlin Wang**: Software; Investigation; Visualization. **Qing Tang**: Validation; Investigation; Writing—review and editing. **Yongde Cai**: Resources; Validation; Investigation. **Siyang Yu**: Validation; Investigation; Writing—review and editing. **Jin Wang**: Resources; Writing—review and editing. **Xiu Yan**: Resources; Writing—review and editing. **Guocai Wang**: Resources; Writing—review and editing. **Peter E Lobie**: Supervision; Writing—review and editing. **Yubo Zhang**: Resources; Supervision; Writing—review and editing. **Xiaoyong Dai**: Conceptualization; Supervision; Funding acquisition; Writing—review and editing. **Shaohua Ma**: Conceptualization; Supervision; Funding acquisition; Project administration; Writing—review and editing.

Source data underlying figure panels in this paper may have individual authorship assigned. Where available, figure panel/source data authorship is listed in the following database record: biostudies:S-SCDT-10_1038-S44321-025-00308-1.

## Disclosure and competing interests statement

The authors declare no competing interests.

# Expanded View Figures

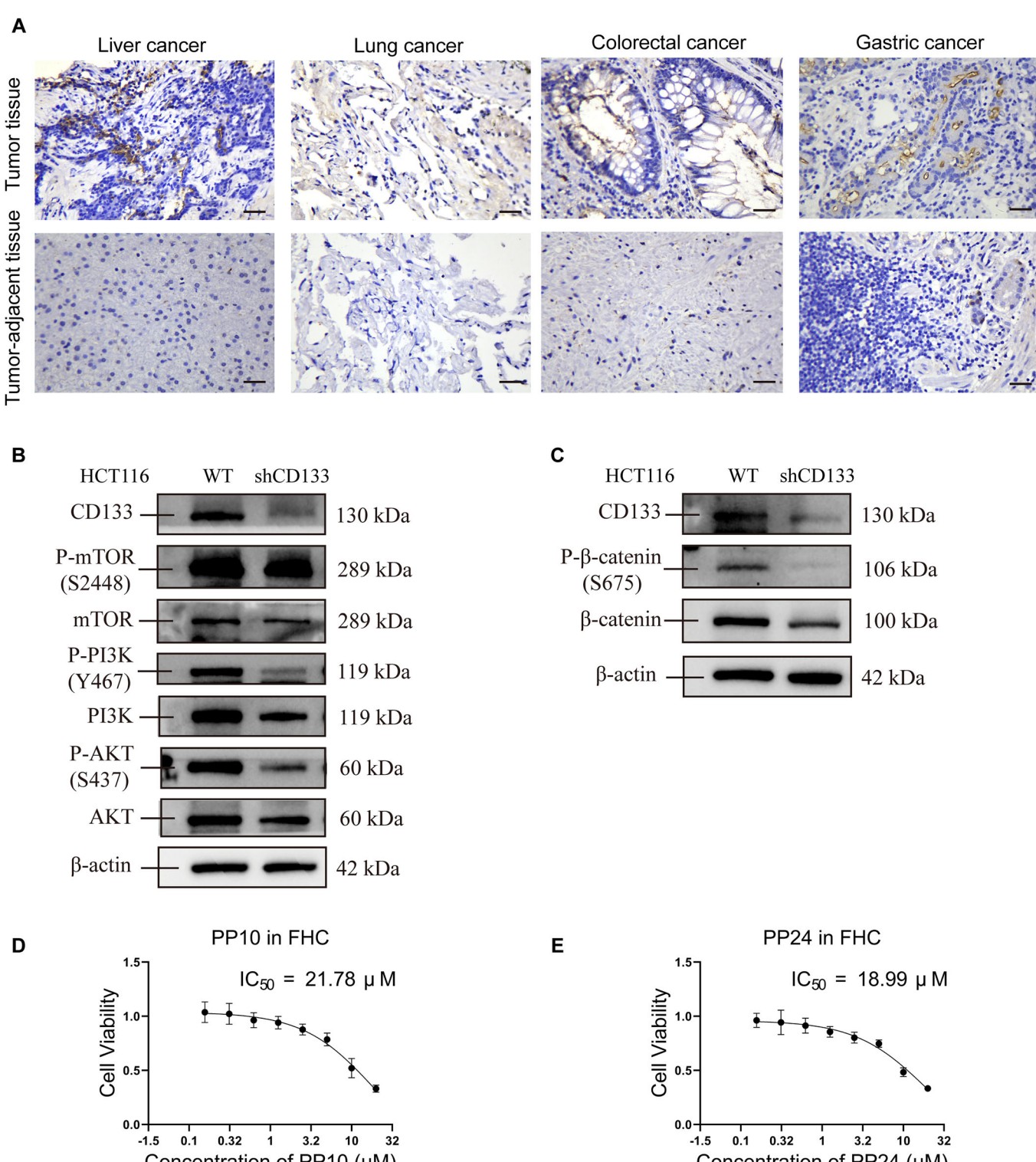

**Figure EV1. Expression level of CD133 in various cancer and cytotoxicity of PP10 and PP24 on normal cell lines.**

(A) The IHC staining of CD133 in various cancer, including liver cancer, lung cancer, colorectal cancer, and gastric cancer. Scale bar, 50 μm. (B, C) Expression level of signaling pathway in WT and shCD133 HCT116 cell lines. (B) PI3K–AKT pathway, (C) Wnt-β-catenin pathway (D, E) Cell viability assay results showing the cytotoxicity of PP10 and PP24 in FHC cells. Data are presented as mean ± SD. *P < 0.05, **P < 0.01, and ***P < 0.001 when compared with control group.

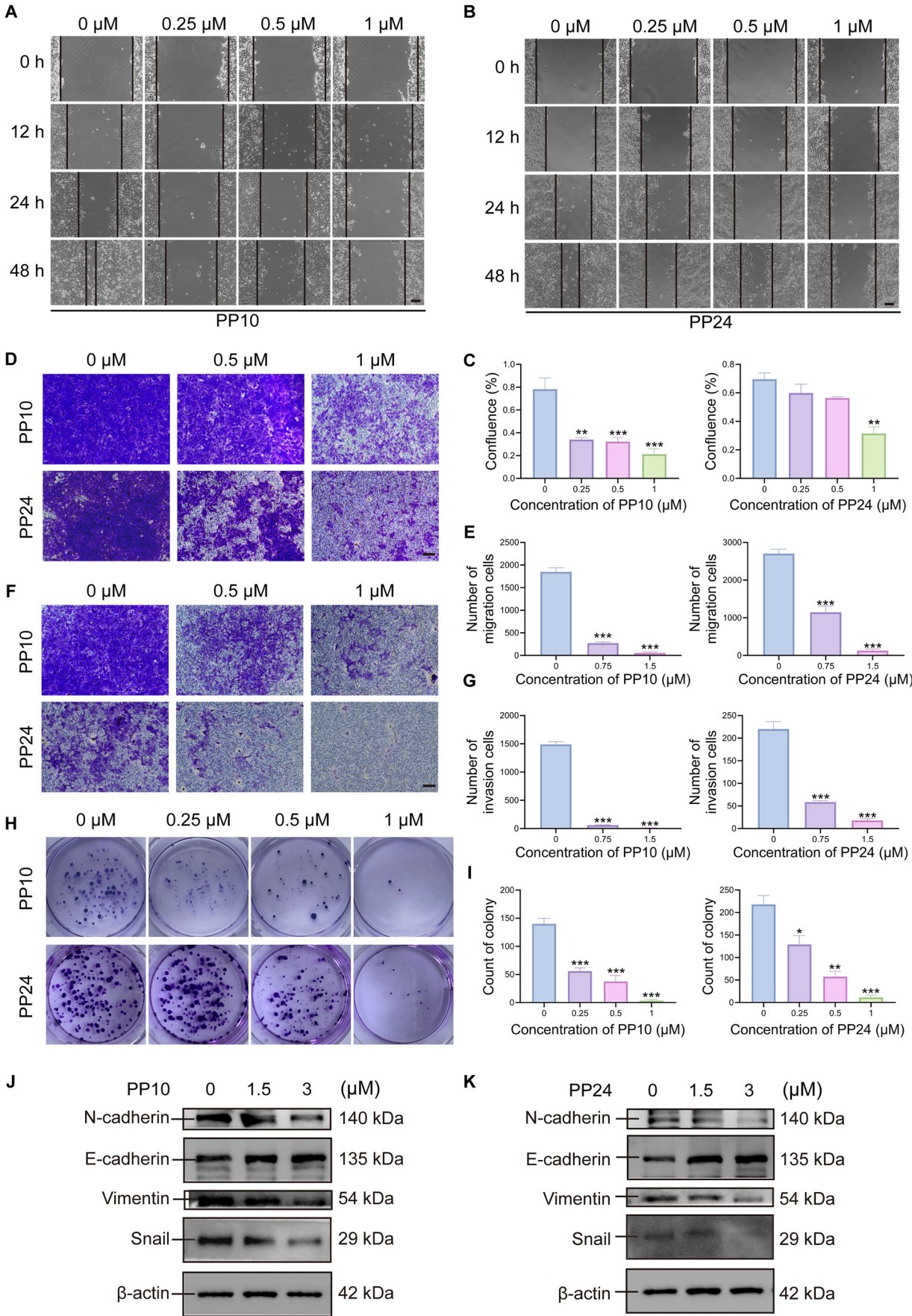

◄ **Figure EV2. PP10 and PP24 inhibit migration, invasion, and colony formation of CRC cells in vitro.**

(A, B) PP10 and PP24 inhibit the wound healing ability of HCT116 cells. (C) Statistics of wound healing confluence in (A, B) ($N = 3$). (D) PP10 and PP24 inhibit the migration ability of HCT116 cells assessed by Transwell assay. (E) Statistics of migrated cell counts in (D) ($N = 3$). (F) PP10 and PP24 suppress the metastasis potential of HCT116 cells in Transwell assays. (G) Statistics of migrated cell counts in (F) ($N = 3$). (H) PP10 and PP24 reduce colony formation ability of HCT116 cells. (I) Statistics of colony numbers formed in (H) ($N = 3$). Western blot analysis of EMT pathway related proteins after treating with PP10 (J) and PP24 (K). Scale bar, 50 μm. one-way ANOVA test. Exact $P$ value are presented in Appendix Table S2. Data are presented as mean ± SD. *$P < 0.05$, **$P < 0.01$, and ***$P < 0.001$ when compared with control group.

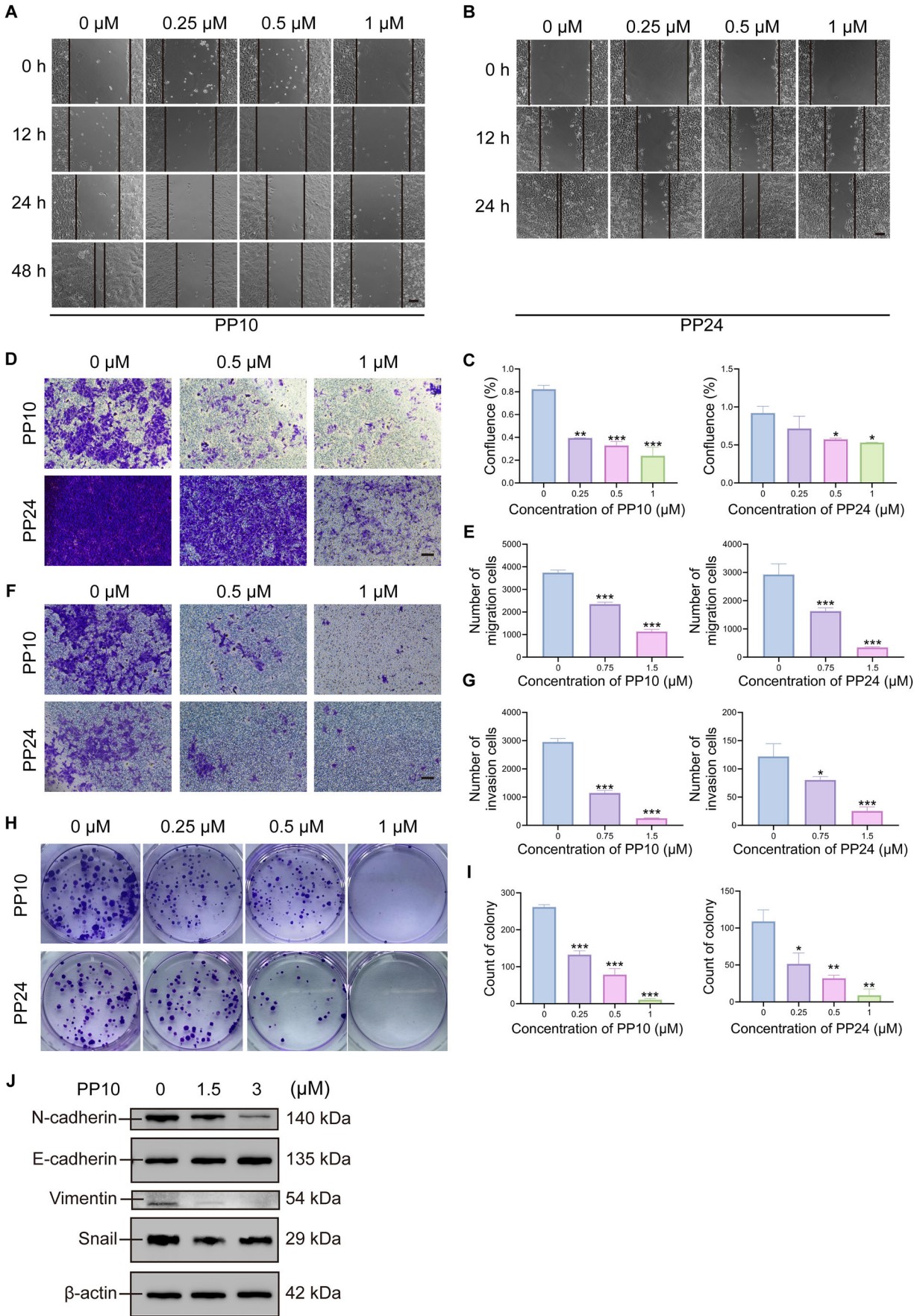

◀ **Figure EV3. PP10 and PP24 inhibit migration, invasion, and colony formation of CRC cells in vitro.**

(A, B) PP10 and PP24 inhibit the wound healing ability of DLD1 cells. (C) Statistics of wound healing confluence in (A, B) ($N = 3$). (D) PP10 and PP24 inhibit the migration ability of DLD1 cells assessed by Transwell assay. (E) Statistics of migrated cell counts in (D) ($N = 3$). (F) PP10 and PP24 suppress the metastasis potential of DLD1 cells in Transwell assays. (G) Statistics of migrated cell counts in (F) ($N = 3$). (H) PP10 and PP24 reduce colony formation ability of DLD1 cells. (I) Statistics of colony numbers formed in (H) ($N = 3$). Western blot analysis of EMT pathway related proteins after treating with PP10 (J). Scale bar, 50 μm. one-way ANOVA test. Exact $P$ value are presented in Appendix Table S2. Data are presented as mean ± SD. $*P < 0.05$, $**P < 0.01$, and $***P < 0.001$, when compared with control group.

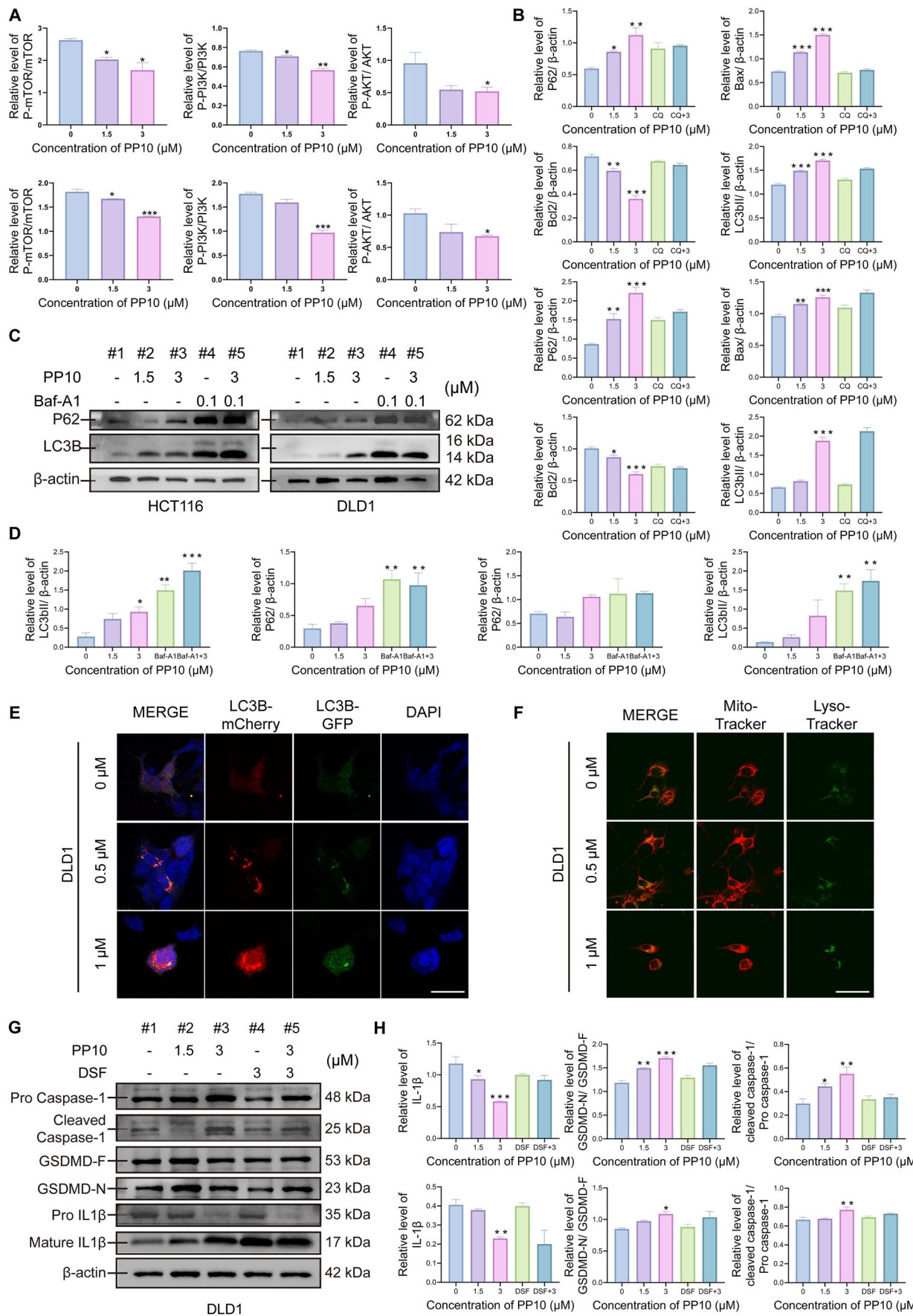

◀ **Figure EV4. Statistical analysis of protein expression level.**

(A) Quantification of expression levels normalized to β-actin in Fig. 4H, I ($N = 3$). (B) Quantification of relative expression levels for the mitophagy pathway to β-actin in Fig. 5C,D ($N = 3$). (C) Western blot analysis of mitophagy-related proteins in HCT116 and DLD1 cells treated with PP10 and Baf-A1. (D) Quantification of relative expression levels for the mitophagy pathway to β-actin in (D) ($N = 3$). (E) Microscopic observation of mCherry-GFP-LC3B infected DLD1 cells, showing autophagosome-lysosome fusion following PP10 treatment. Scale bar, 50 μm. (F) Microscopic observation of mitochondria and lysosome co-localization in DLD1 cells using Mito/Lyso-Tracker probes after PP10 treatment. Scale bar, 50 μm. (G) Western blot analysis of pyroptosis-related proteins in DLD1 cells treated with PP10. (H) Quantification of relative expression levels for pyroptosis pathway to β-actin in Fig. 5I, Fig. EV4G ($N = 3$). one-way ANOVA test. Exact P value are presented in Appendix Table S2. Data are presented as mean ± SD. $*P < 0.05$, $**P < 0.01$, and $***P < 0.001$, when compared with control group.

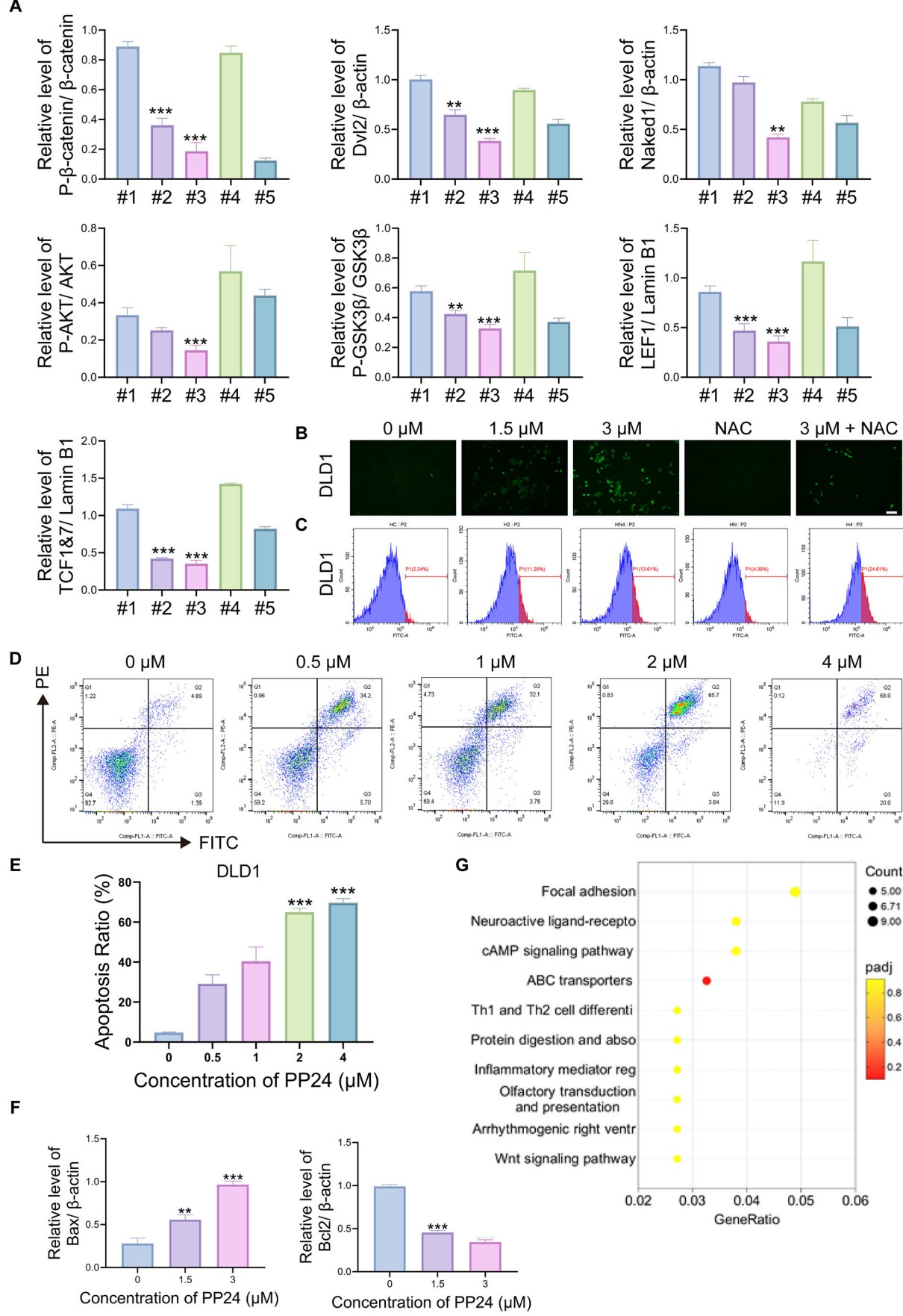

◀ **Figure EV5. Statistical analysis of protein expression level and detection of ROS.**

(A) Quantification of relative expression levels to β-actin and Lamin B1 ($N = 3$) in Fig. 6C. (B) Microscopic observation of ROS levels in DLD1 treated with PP24 and/or NAC using DCFH-DA probe. (C) Flow cytometry results showing ROS detection after treatment with PP24 and NAC in DLD1 cells. (D, E) Detection of apoptosis of DLD1 cells after treatment of PP24 by flow cytometry. (F) Quantification of relative expression levels to β-actin in Fig. 6F ($N = 3$). (G) KEGG pathway enrichment analysis showing the functional pathways affected by DEGs in DLD1 cells following PP24 treatment. One-way ANOVA test. Exact $P$ value are presented in Appendix Table S2. Data are presented as mean ± SD. *$P < 0.05$, **$P < 0.01$, and ***$P < 0.001$, when compared with control group.

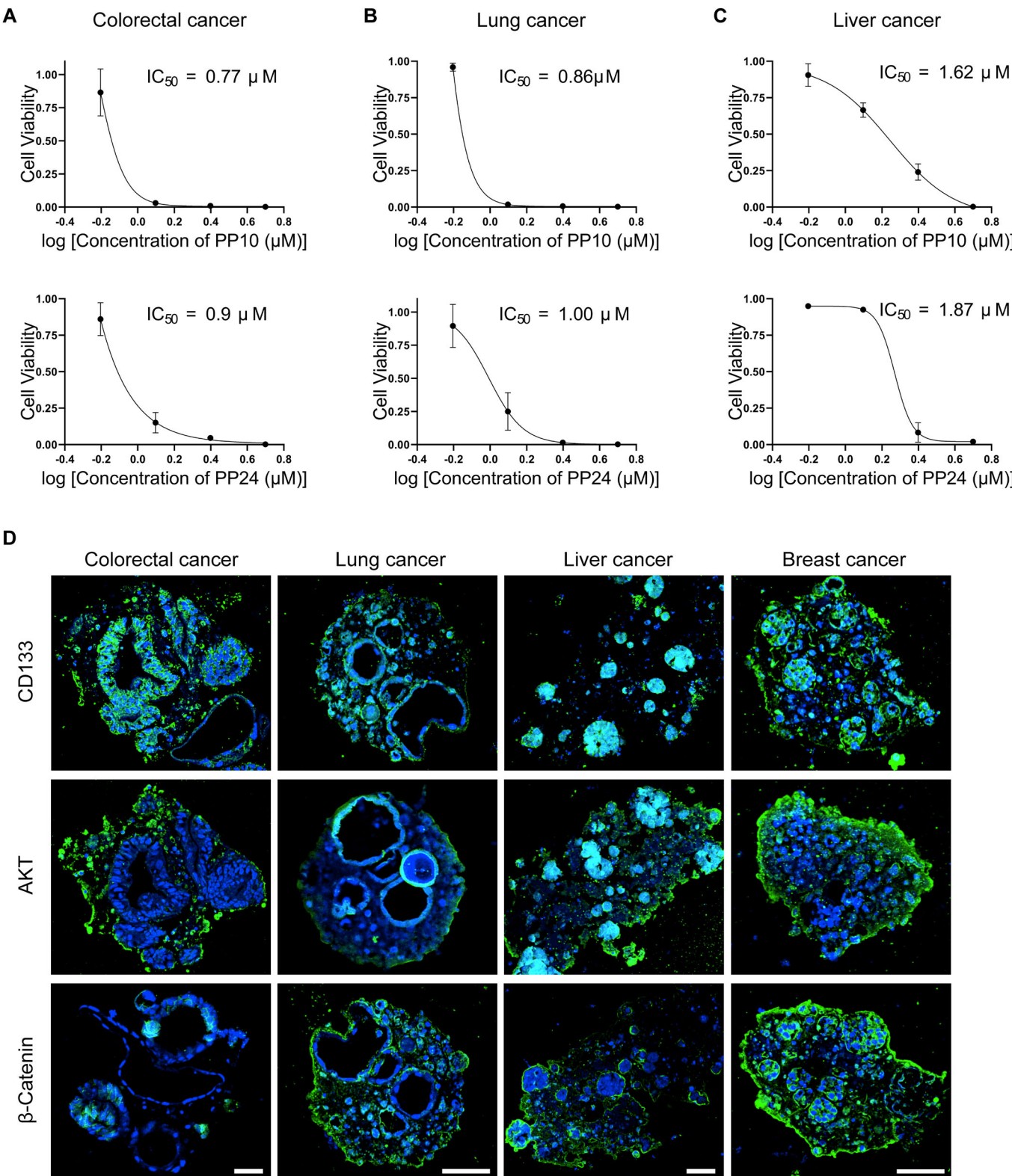

**Figure EV6.  PP10 and PP24 inhibit viability of pan-cancer organoids.**

PP10 and PP24 inhibited the cell viability of colorectal cancer organoids (**A**), lung cancer organoids (**B**) and liver cancer organoids (**C**) ($N = 3$). (**D**) Expression level of CD133 in pan-cancer patient-derived organoids, namely, Colorectal cancer, Lung cancer, Liver cancer, Thyroid cancer, and Breast cancer. Scale bar $= 50$ μm. one-way ANOVA test. Data are presented as mean ± SD. *$P < 0.05$, **$P < 0.01$, and ***$P < 0.001$, when compared with control group.

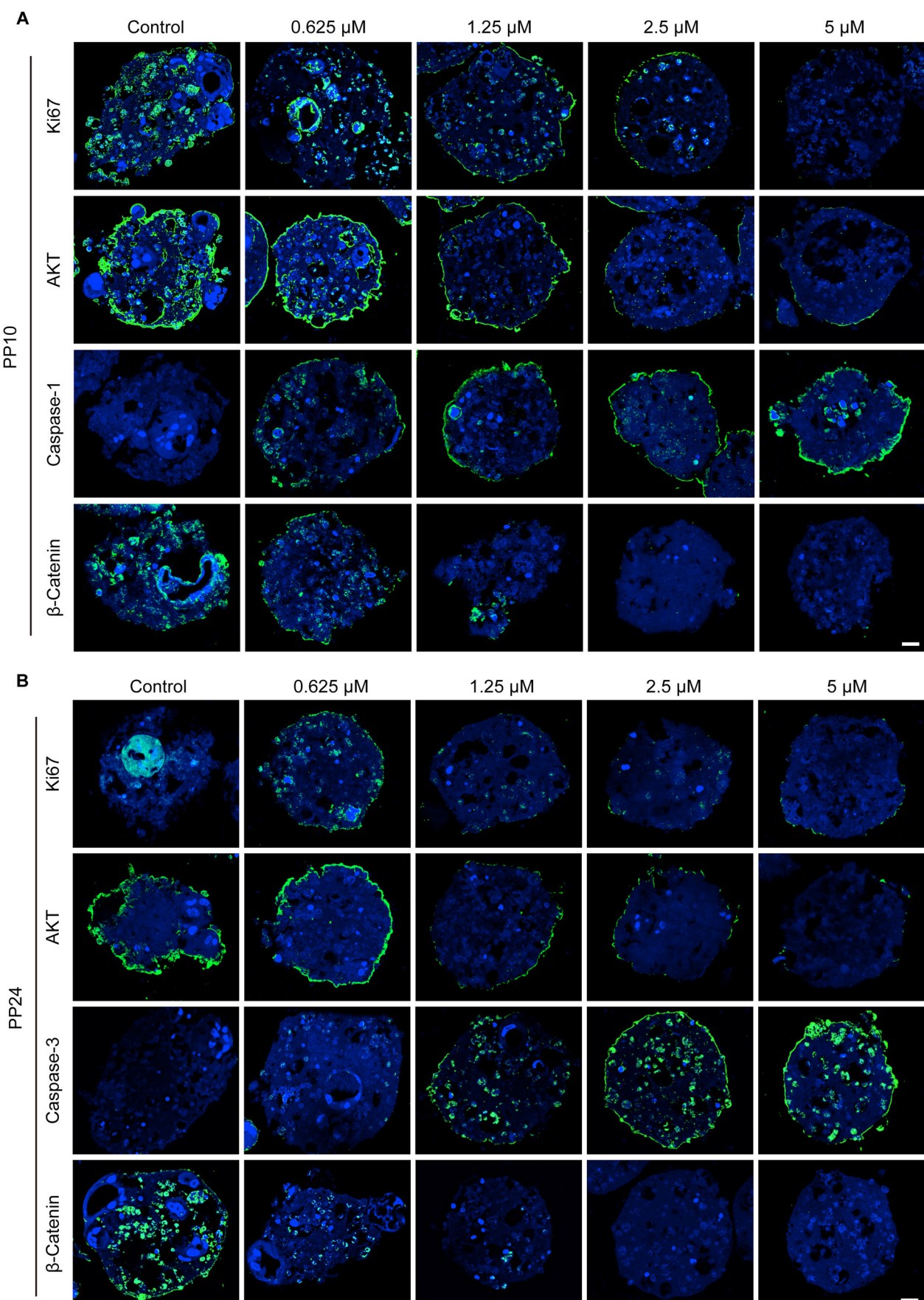

◄ **Figure EV7. Immunofluorescence staining of organoid.**

(A) Immunofluorescence staining of Ki67, AKT, Caspase-1, and β-catenin in CRC organoids after treatment with PP10. (B) Immunofluorescence staining of frozen sections of CRC organoids after treatment with PP24 on Ki67, AKT, Caspase-3, and β-Catenin. Scale bar, 50 μm.

