## [Peer Review File · EMBO Molecular Medicine]

AI-Identified CD133-Targeting Natural Compounds Demonstrate Differential Anti-Tumor Effects and Mechanisms in Pan-cancer Models

Yibo Hou, Zixian Wang, Wenlin Wang, Qing Tang, Yongde Cai, Siyang Yu, Jin Wang, Xiu Yan, Guocai Wang, Peter Lobie, Yubo Zhang, Xiaoyong Dai, and Shaohua Ma

Corresponding authors: Shaohua Ma (ma.shaohua@sz.tsinghua.edu.cn) , Xiaoyong Dai (daixy18@jnu.edu.cn), Yubo Zhang (ybzhang99@jnu.edu.cn)

Review Timeline:

Submission Date:	28th Feb 25
Editorial Decision:	14th Apr 25
Revision Received:	19th Jun 25
Editorial Decision:	17th Jul 25
Revision Received:	4th Aug 25
Accepted:	28th Aug 25

Editor: Zeljko Durdevic

Transaction Report:

14th Apr 2025

Dear Dr. Ma,

Thank you for the submission of your manuscript to EMBO Molecular Medicine, and please accept my apologies for the unusual delay in getting back to you. We have now received feedback from two of the three reviewers who agreed to evaluate your manuscript. As the referee #3 will unfortunately not be able to return his/her report in a timely manner, and given that both reviewers provide very similar recommendations, we prefer to make a decision now in order to avoid further delay in the process. Should referee #3 provide a report, we will send it to you, with the understanding that we will not ask for an additional revision.

As you will see from their reports pasted below, both referees recognize potential interest of the manuscript but also raise serious and partially overlapping concerns that should be addressed in a major revision. If you would like to discuss further the points raised by the referees, I am available to do so via email or video. Let me know if you are interested in this option.

We would welcome the submission of a revised version within three months for further consideration. Please let us know if you require longer to complete the revision.

I look forward to receiving your revised manuscript.

Yours sincerely,

Zeljko Durdevic

We require:

- 1) A .docx formatted version of the manuscript text (including legends for main figures, EV figures and tables). Please make sure that the changes are highlighted to be clearly visible.
- 2) Individual production quality figure files as .eps, .tif, .jpg (one file per figure). For guidance, download the 'Figure Guide PDF': (<https://www.embopress.org/page/journal/17574684/authorguide#figureformat>).
- 3) A .docx formatted letter INCLUDING the reviewers' reports and your detailed point-by-point responses to their comments. As part of the EMBO Press transparent editorial process, the point-by-point response is part of the Review Process File (RPF), which will be published alongside your paper.
- 4) A complete author checklist, which you can download from our author guidelines (<https://www.embopress.org/page/journal/17574684/authorguide#submissionofrevisions>). Please insert information in the checklist that is also reflected in the manuscript. The completed author checklist will also be part of the RPF.

6) It is mandatory to include a 'Data Availability' section after the Materials and Methods. Before submitting your revision, primary datasets produced in this study need to be deposited in an appropriate public database, and the accession numbers and database listed under 'Data Availability'. Please remember to provide a reviewer password if the datasets are not yet public (see <https://www.embopress.org/page/journal/17574684/authorguide#dataavailability>).

.

- the medical issue you are addressing,

- the results obtained and

- their clinical impact.

12) Author contributions: You will be asked to provide CRediT (Contributor Role Taxonomy) terms in the submission system. These replace a narrative author contribution section in the manuscript.

13) A Conflict of Interest statement should be provided in the main text.

14) Every published paper now includes a 'Synopsis' to further enhance discoverability. Synopses are displayed on the journal webpage and are freely accessible to all readers. They include a short stand first (maximum of 300 characters, including space) as well as 2-5 one-sentences bullet points that summarizes the paper. Please write the bullet points to summarize the key NEW findings. They should be designed to be complementary to the abstract - i.e. not repeat the same text. We encourage inclusion of key acronyms and quantitative information (maximum of 30 words / bullet point). Please use the passive voice. Please attach these in a separate file or send them by email, we will incorporate them accordingly.

15) Include a Reagents and Tools Table as part of the Methods section, which can be downloaded from our author guidelines (<https://www.embopress.org/page/journal/17574684/authorguide#structuredmethods>)

***** Reviewer's comments *****

Referee #1 (Comments on Novelty/Model System for Author):

The topic is overall interesting, although the work focuses on applying the existing AI tools, but not really developing some new tools, to facilitate the identification of new compounds. The story could be more interesting if the authors could work out the mechanism on how the binding of PP10 and PP24 to CD133 could lead to distinct effects on the signalling pathways structurally. More detailed comments appended below.

Referee #1 (Remarks for Author):

In this work, the authors used the BindingDB to train a Transformer-based compound-protein interaction deep learning model. Using the algorithm, 29 compounds were identified to be potentially interacting with the CD133, which is expressed in various types of cancer cells compared to the adjacent normal tissue cells. Among the compounds, PP10 and PP24 exhibit significant cytotoxic activities. The authors then elucidated the mechanisms of the two compounds and found that PP10 exerts anti-tumour effect via inhibiting the PI3K/AKT/mTOR pathway while the PP24 exerts its effect via inhibiting Wnt/ β -catenin pathway. Finally, the authors validated the anti-tumour effect of PP10 and PP24 using patient derived organoids and subcutaneous xenograft mouse models. Several comments are appended below:

1. The authors used the Byte Pair Encoding which represents amino acids with binary vectors only. Would other encoding methods like physiochemical properties encoding or evolution-based encoding improve the performance of the model?
2. How the direct inhibition of CD133 would affect the downstream PI3K or Wnt signalling? Does CD133 involve in these pathways for cell signalling? What is the mechanism?
3. It would be commendable for the authors to work out the mechanism and elaborate more on how the binding of PP10 and PP24 to CD133 could lead to distinct effects on the signalling pathways structurally.
4. Would PP10 and PP24 exerts cytotoxicity to CD133 negative cancer cells?
5. According to the KEGG analysis for the treatment of PP24 (Fig.6b), why estrogen signalling pathway is highly enriched in HCT116 cell line? How about the KEGG analysis result for DLD1 cell line?

Referee #2 (Comments on Novelty/Model System for Author):

The drug validation was not conducted in models that would enable the assessment of compound selectivity and CD133 specificity. Consequently, the study lacks biological evidence to support the validity of the AI-based drug discovery approach. These concerns are outlined in the comments to the authors, along with suggestions for addressing them.

Referee #2 (Remarks for Author):

This manuscript presents an interesting AI-guided screening strategy to identify natural compounds targeting prominin-1 (CD133), a protein commonly associated with cancer stem cells and poor cancer survival. Given the limited options to therapeutically target this protein so far, the incorporation of deep learning into early-phase drug discovery presents a notable strength of this work. The authors screened a library of 1,123 traditional Chinese medicine-derived molecules in silico to identify

29 compounds potentially interacting with CD133. From these, two steroidal saponins-polyphyllin V (PP10) and polyphyllin H (PP24)-were selected and validated in vitro and in vivo. This study demonstrated the antitumor effects of PP10 and PP24 in various cancer models and described potential molecular mechanisms of action. Nonetheless, as detailed in the comments below, the relevance of these findings and the utility of the reported AI-based approach for identifying CD133-targeted therapeutics remain uncertain due to limited biological evidence on compound selectivity and CD133 specificity, which undermines the conclusions and impact of the study.

First, the rationale for selecting the most potent drugs is not clearly described. In the first in vitro validation of the hits, CL6 showed the lowest IC50 in CRC HCT116 cells. However, it was not further analyzed in the panel of cell lines (Extended Data Fig. 2). Although the top 6 identified compounds are not described in full names (except for polyphyllin V and polyphyllin H), Refs 39,49 used to support that CL6 has been already studied (Page 6, first paragraph) do not contain any information on CL6, as these Refs focus on polyphyllin II and VII. Since CL6 showed the most potent antiproliferative effects of the identified candidates, it is important to validate CL6's CD133-specific antitumor activities. Otherwise, the authors should provide a strong rationale for excluding CL6, the top hit in silico and in vitro, from this AI-based discovery and validation of CD133-targeting drugs.

A major concern is the lack of functional validation of the selective CD133-dependent activities of the studied drugs, PP10 and PP24. As natural compounds often show complex effects on multiple cellular targets, the observed activities may be independent of their binding to CD133. However, there are no mechanistic experiments to support that the described antitumor properties and different effects on cell signaling, mitochondrial physiology, and cell death are caused by blocking CD133 activity. Several related comments that should be addressed are listed below:

1) Cellular thermal shift assay (CETSA) experiments represent crucial biological evidence of direct binding of the drugs to CD133. However, the results and analysis presented are not technically sound: (i) Loading appears uneven, as 45°C samples from both HCT116 and DLD1 show less CD133 in the presence of PP24 than in respective controls. This either indicates that there was a significant technical variance in loading or that the drug decreased CD133 stability. (ii) Barely detectable bands in the blots translate only into approx. 2-fold decrease in densitometry (e.g., PP10/60°C in HCT116 - Fig. 2i vs Fig. 2k; PP10 samples in DLD1 - Fig. 2i vs. Extended Data Fig. 2c). (iii) In contrast with these technical discrepancies, error bars in all related graphs show extremely low SD, which is very uncommon for immunoblotting methods. Providing the blots from the two other biological replicates in Extended Data Fig. 2 would strengthen this CETSA analysis and the conclusions made.

2) Expression of CD133 was not characterized and compared among the cell lines used for drug validation (Extended Data Fig. 2), which prevents interpretation of CD133-specific cytotoxicity of the drugs. Both HCCLM3 and A549 are known to express CD133, but they showed significantly lower sensitivity to PP10 and PP24. Conversely, CD133 has been previously found absent or minimally expressed in DLD1 (PMID: 19738050; PMID: 22895640; PMID: 26002465), yet DLD1 cells showed similarly high sensitivity to PP10 and PP24 as HTC116 CD133-overexpressing cells. To validate PP10 and PP24 as potential drugs for CD133-targeted therapies, it is necessary to characterize CD133 protein expression and include CRC models with high as well as low/absent CD133 levels to test for the drug selectivity against CD133-overexpressing cells. Otherwise, the observed effects could be attributed to other off-target activities of PP10 and PP24.

3) The authors performed several experiments to elucidate the mechanism of antitumor effects of PP10 and PP24. However, these experiments lack essential controls - either using CD133 knockdown or interaction mutants (e.g., p85-binding deficient Y828F CD133) - to determine whether removing the target or preventing its interaction with signaling molecules diminishes the effects of PP10/PP24. Without this evidence, it is speculative to claim that the observed changes in cell signaling and other processes are due to the binding of the drugs to CD133 ("Our results fully demonstrate that PP10 and PP24 can specifically bind to CD133 to achieve pan-cancer therapy"). In fact, the potent antitumor activity and divergent mechanisms of action of PP10 vs. PP24 could be related to their different effects on other molecules independent of CD133.

Finally, the choice of autophagy inhibitor could flaw the conclusions about PP10-induced autophagy/mitophagy. Chloroquine (CQ) is a suboptimal autophagy inhibitor with known Golgi-related effects, which can paradoxically induce autophagy (PMID: 29940786). This warrants caution when interpreting results obtained by blocking autophagy with this drug. To measure autophagy flux, bafilomycin A1 should be used to inhibit autophagosome-lysosome fusion and cargo sequestration (PMID: 33634751). Importantly, results shown in Fig. 5c,d may indicate that PP10 inhibits rather than promotes autophagy flux. LC3-II forms were accumulated after PP10 treatment alone, and inhibiting the autophagy did not increase their levels, suggesting that PP10 already blocked autophagy in terminal stages (see also 2b in PMID: 33634751). However, using CQ, which clearly could not inhibit autophagy in the case of DLD1 untreated cells (Fig. 5d, lane #1 vs #4), prevents any interpretation of the results. Similarly, microscopy analysis of mitophagy induction cannot be properly interpreted without comparing PP10-treated cells in the presence and absence of autophagy inhibition by bafilomycin A1. The accumulation of mitochondria in lysosomes may, in fact, again indicate inhibited autophagy at the final stages by PP10 (not increased autophagy). Confocal microscopy results should then be supported by quantitative image analysis across multiple fields of view per biological replicate.

Dear Editors and Reviewers,

We are grateful to reviewers for taking the time to review our manuscript and give us their valuable comments. We have considered all the comments and have made appropriate changes to the manuscript. Our point-by-point response appears below in blue. The changes made to the main manuscript file were highlighted.

Referee #1:

In this work, the authors used the BindingDB to train a Transformer-based compound-protein interaction deep learning model. Using the algorithm, 29 compounds were identified to be potentially interacting with the CD133, which is expressed in various types of cancer cells compared to the adjacent normal tissue cells. Among the compounds, PP10 and PP24 exhibit significant cytotoxic activities. The authors then elucidated the mechanisms of the two compounds and found that PP10 exerts anti-tumour effect via inhibiting the PI3K/AKT/mTOR pathway while the PP24 exerts its effect via inhibiting Wnt/ β -catenin pathway. Finally, the authors validated the anti-tumour effect of PP10 and PP24 using patient derived organoids and subcutaneous xenograft mouse models. Several comments are appended below:

Q1: The authors used the Byte Pair Encoding which represents amino acids with binary vectors only. Would other encoding methods like physiochemical properties encoding or evolution-based encoding improve the performance of the model?

Answer: We deeply appreciate the reviewer's insightful suggestion regarding sequence representation methods, which can prompt us to re-evaluate our encoding strategy from a biological perspective. While BPE is advantageous for general sequence tokenization, we recognize that domain-specific encodings can better capture the biophysical and evolutionary constraints of proteins. In response, we implemented a more biologically grounded approach using ProteinBERT, which leverages the unique physicochemical characteristics of proteins and functional annotations from Gene Ontology terms¹. Our updated results show this approach

maintains competitive performance (with PP-10 and PP-24 still in the top 30 rankings) and offers more biologically interpretable representations, aligning better with protein science principles.

a	Rank	Score	Reference Number	b	Rank	Score	Reference Number
	1	95.5%	CL-2		1	85.4%	BM-29
	2	96.3%	CL-4		2	85.2%	CL-3
	3	98.0%	PP-9		3	85.1%	PP-29
	4	97.0%	CL-5		4	84.5%	ZMC-25
	5	98.0%	PP-22		5	84.5%	PP-30
	6	97.9%	PP-18		6	84.2%	BM-51
	7	98.6%	CL-9		7	83.8%	ZMC-21
	8	97.7%	PP-10		8	83.8%	PP-5
	9	95.8%	PP-27		9	83.8%	PP-9
	10	92.0%	HYZ-3		10	83.7%	PP-28
	11	95.4%	FYC-7		11	83.6%	LLY-1
	12	95.0%	SC-14		12	83.3%	PP-10
	13	95.0%	PQJ-22		13	83.2%	PP-19
	14	95.4%	CL-6		14	83.2%	PP-24
	15	81.7%	DCN-3		15	83.1%	PP-6
	16	89.4%	PP-20		16	83.0%	CL-6
	17	85.8%	PP-23		17	83.0%	PP-11
	18	94.4%	PP-28		18	82.9%	PP-26
	19	94.4%	SC-13		19	82.8%	PP-46
	20	95.5%	SC-15		20	82.7%	PP-23
	21	72.8%	SPH-16		21	82.7%	CL-9
	22	91.5%	HYZ-4		22	82.5%	PP-16
	23	91.9%	FYC-1		23	82.5%	PP-20
	24	77.1%	FYC-6		24	82.4%	PP-27
	25	88.1%	CL-18		25	82.3%	PP-3
	26	94.3%	PP-6		26	82.0%	PP-17
	27	94.5%	PP-8		27	81.8%	PP-25
	28	94.7%	PP-19		28	81.8%	LLY-2
	29	91.5%	PP-24		29	81.8%	PP-22
	30	87.7%	(-)-DCN-15		30	81.8%	PP-14

Fig. 1 AI-based drug screening results targeting CD133. (a) Drug screening results base on Byte Pair Encoding (The previous encoding method). (b) Drug screening results base on ProteinBERT (updated encoding method).

Q2: How the direct inhibition of CD133 would affect the downstream PI3K or Wnt signalling? Does CD133 involve in these pathways for cell signalling? What is the mechanism?

Answer: Thanks for your suggestion. To clarify the effects of CD133 on downstream PI3K and Wnt signaling pathways, as well as to investigate the impacts of PP10 and PP24 on CD133-negative cell lines, we constructed a CD133 knockdown cell line using HCT116 cells. Lentivirus-mediated transfection was employed to introduce the constructed CD133 shRNA plasmid into the target cells, and cells were screened with puromycin to obtain stable cell line. The Western blot results showed a significant decrease in the shCD133 group when compared to the wild type (WT) group, achieving sustained knockdown of the target protein. Further experiments showed that

knockdown CD133 significantly blocked both the PI3K and Wnt signalling by decreasing the phosphorylation level of mTOR, PI3K, AKT and β -catenin. Thus, inhibiting CD133 with PP10 or PP24 could suppress the **downstream PI3K or Wnt signalling** (Fig. 2a, b).

Many studies have demonstrated that CD133 is associated with the PI3K/AKT signaling pathway^{2,3}. The phosphorylation of CD133 is a key step in mediating PI3K/AKT signaling. Specifically, CD133 can recruit the p85 regulatory subunit of PI3K to the plasma membrane, thereby activating AKT⁴. Additionally, CD133 can stabilize receptor tyrosine kinases (e.g., EGFR) on the plasma membrane, preventing their endocytosis and sustaining AKT activation⁵. Our results showed that downregulation of CD133 by PP10 or shRNA could inhibit PI3K/AKT/mTOR pathway via decreasing the **downstream protein levels**.

CD133 is also involved in the Wnt/ β -catenin signaling pathway. CD133 can enhance Wnt signaling by increasing β -catenin levels through AKT-mediated GSK3 inhibition, preventing β -catenin from undergoing β -TrCP polyubiquitination and subsequent degradation⁶. In the shCD133 group, the expression levels of AKT and β -catenin were all reduced, a similar effect to that observed in the PP24-treated group. These results indicated that the suppression of the Wnt signaling pathway by shCD133 was related to PP24-induced CD133 inhibition.

Fig. 2 Expression level of signaling pathway in WT and shCD133 HCT116 cell lines. (a) Western blot result of PI3K/AKT pathway in the WT HCT116 and shCD133-HCT116 cells. (b) Western blot result of Wnt pathway in the WT HCT116 and shCD133-HCT116 cells.

Q3: It would be commendable for the authors to work out the mechanism and elaborate more on how the binding of PP10 and PP24 to CD133 could lead to distinct effects on the signalling pathways structurally.

Answer: We appreciate your suggestion. Based on molecular docking results, we found that PP10 and PP24 bind to different regions of CD133, which may result in different inhibitory effects on downstream pathways. Molecular docking results revealed that PP10 bound to CD133 through hydrophobic interactions at residues ALA-2, LEU-5, PHE-62, TRP-786, PHE-787, and LYS-791, and formed hydrogen bonds at LEU-3 and PHE-795. Similarly, PP24 bound to CD133 via hydrophobic interactions at LEU-76, LEU-80, LEU-505, ILE-506, and PHE-516, and formed hydrogen bonds at PRO-72 and GLU-97. RNA sequencing results further helped us explore the distinct effects caused by PP10 and PP24. For PP10, several genes in the PI3K/AKT pathway were downregulated, including ITGA, RTK, PP2A, and SGK. ITGA and RTK, which are associated with poor progression of CRC and can activate

the PI3K pathway, showed a positive correlation with CD133^{7,8}. The reduction of ITGA and RTK in the PP10 treatment group may suggest an interaction between CD133 and ITGA or RTK (Fig. 3). In the PP24 treatment group, Wnt, Wnt11, and BAMBI in the Wnt pathway were downregulated. Wnt and Wnt11 are key proteins in the pathway, while BAMBI plays an important role by interacting with LRP5/6 to enhance Wnt pathway activity⁹. This may imply a potential interaction between CD133 and BAMBI (Fig. 4). Overall, PP10 and PP24 exhibit different interaction modes with CD133, leading to distinct downstream responses and differing effects on the two signaling pathways.

Fig. 3 Expression level of PI3K/AKT signaling pathway in PP10 treated group of KEGG analysis.

Fig. 4 Expression level of Wnt signaling pathway in PP24 treated group of KEGG analysis.

Q4: Would PP10 and PP24 exerts cytotoxicity to CD133 negative cancer cells?

Answer: We are grateful for your suggestion. To investigate the cytotoxicity of PP10 and PP24 on CD133-negative cancer cells, we constructed a CD133-knockdown HCT116 cell line and validated the decreased CD133 expression via Western blot. We also selected the human normal colorectal mucosal cell line FHC to evaluate the cytotoxicity of PP10 and PP24 on normal cells. CCK8 assays showed a significant increase in the IC₅₀ concentration of both FHC and shCD133-HCT116 cell lines, from approximately 1 μM to 18.99-28.64 μM. For PP10, the IC₅₀ is 21.78 μM for FHC cell and 28.64 μM for shCD113-HCT116 cell. For PP24, the IC₅₀ is 18.99 μM for FHC cell and 25.39 μM for shCD113-HCT116 cell (Fig. 5). Firstly, the lower cytotoxicity in the normal cell line suggests that PP10 and PP24 have relatively high safety profiles. Secondly, their reduced cytotoxicity in CD133-low-expressing cancer cells indicates that these two compounds may specifically target CD133. Overall, these results highlight the selective targeting ability of PP10 and PP24 toward CD133 overexpressed cancer cells, while exerting lower cytotoxicity on CD133-negative

cancer cells and normal cells.

Fig. 5 Cell viability of FHC and shCD133-HCT116 cells treated by PP10 and PP24 (a, b) CCK8 assay results showing the cytotoxicity of PP10 and PP24 in FHC cells. (c, d) CCK8 assay results showing the cytotoxicity of PP10 and PP24 in shCD133-HCT116 cells.

Q5: According to the KEGG analysis for the treatment of PP24 (Fig.6b), why estrogen signalling pathway is highly enriched in HCT116 cell line? How about the KEGG analysis result for DLD1 cell line?

Answer: Thankful for your meaningful suggestion. According to the KEGG analysis for PP24 treatment, the estrogen signaling pathway was found to be enriched in the PP24-treated HCT116 cell line compared to the control group, but with decreased activity. The ER pathway is associated with Wnt pathway in CRC. ER signaling exhibits dual effects in CRC progression. ER α can promote CRC cell proliferation, while ER β may suppress it¹⁰. High ER α expression is linked to a higher risk of CRC liver metastasis¹¹. In ER α -positive/HER2-negative breast cancer, tumors with high CD133 expression exhibit obvious cancer stem cell (CSC) characteristics. CD133 is closely associated with the activation of Wnt/ β -catenin and Notch pathways, which

are also involved in ER α -regulated tumor biology¹². Moreover, ER α and β -catenin can co-localize to promote downstream genes transcription¹³. In the PP24-treated group, several genes were downregulated, including KRT38, KRT35, FOS, KRT16P6, KRT37, EGFR-AS1, and ESR1. Herein, KRT38, KRT35, KRT16P6, and KRT37 belong to the cytokeratin family and are involved in cytoskeleton formation. Their downregulation may indirectly influence the intracellular signal transduction microenvironment, including Wnt signal transmission efficiency and direction, by affecting cytoskeleton stability and arrangement. FOS, a transcription factor, can interact with members of the JUN family to form the AP-1 complex and participate in regulating the Wnt signaling pathway¹⁴. ESR1 encodes ER α , and abnormal activation of the Wnt signaling pathway may collaborate with ER signaling dysregulation to drive tumor cell malignant transformation. Thus, during tumorigenesis, ER α encoded by ESR1 may interact with the Wnt signaling pathway. However, the specific details and implications of this finding would need further investigation and analysis.

In our research, we primarily focused on the Wnt pathway due to the limited evidence available on the role of ER signaling in CRC progression and its interaction with the Wnt pathway. This focus allowed us to better understand the mechanisms underlying the effects of PP24 on CRC cells.

Referee #2 (Comments on Novelty/Model System for Author):

The drug validation was not conducted in models that would enable the assessment of compound selectivity and CD133 specificity. Consequently, the study lacks biological evidence to support the validity of the AI-based drug discovery approach. These concerns are outlined in the comments to the authors, along with suggestions for addressing them.

Referee #2 (Remarks for Author):

This manuscript presents an interesting AI-guided screening strategy to identify natural compounds targeting prominin-1 (CD133), a protein commonly

associated with cancer stem cells and poor cancer survival. Given the limited options to therapeutically target this protein so far, the incorporation of deep learning into early-phase drug discovery presents a notable strength of this work. The authors screened a library of 1,123 traditional Chinese medicine-derived molecules in silico to identify 29 compounds potentially interacting with CD133. From these, two steroidal saponins-polyphyllin V (PP10) and polyphyllin H (PP24)-were selected and validated in vitro and in vivo. This study demonstrated the antitumor effects of PP10 and PP24 in various cancer models and described potential molecular mechanisms of action. Nonetheless, as detailed in the comments below, the relevance of these findings and the utility of the reported AI-based approach for identifying CD133-targeted therapeutics remain uncertain due to limited biological evidence on compound selectivity and CD133 specificity, which undermines the conclusions and impact of the study.

Answer: Thank you for your positive comments and valuable suggestions on our manuscript. We highly appreciate your recognition of our AI-guided screening strategy for identifying natural compounds targeting CD133 and your useful comments regarding the compound selectivity and CD133 specificity. In response to your concerns, we have carefully evaluated the feasibility and relevance of your suggestions and have already carried out some additional experiments to provide more solid evidence of the targeting effects and biological effects of the compounds. We expect that these supplementary data will address the concerns you raised and strengthen the conclusions of our study.

Q1: First, the rationale for selecting the most potent drugs is not clearly described. In the first in vitro validation of the hits, CL6 showed the lowest IC₅₀ in CRC HCT116 cells. However, it was not further analyzed in the panel of cell lines (Extended Data Fig. 2). Although the top 6 identified compounds are not described in full names (except for polyphyllin V and polyphyllin H), Refs 39,49 used to support that CL6 has been already studied (Page 6, first paragraph) do not contain any information on CL6, as these Refs focus on polyphyllin II and

VII. Since CL6 showed the most potent antiproliferative effects of the identified candidates, it is important to validate CL6's CD133-specific antitumor activities. Otherwise, the authors should provide a strong rationale for excluding CL6, the top hit in silico and in vitro, from this AI-based discovery and validation of CD133-targeting drugs.

Answer: Thank you for your insightful comments. We sincerely apologize for the lack of clarity in marking the correct name of CL6. In our manuscript, CL6 corresponds to polyphyllin II. We have now added the full name of CL6 to avoid any confusion. While Polyphyllin II (CL6) demonstrated excellent cytotoxicity against cancer cell lines, there are several important reasons that hindered its further investigation in our study:

Firstly, the extraction efficiency of Polyphyllin II in our lab was initially quite low, which could not meet the demands of functional study. *Paris polyphylla*, the plant source of Polyphyllin II, has significant medicinal and economic value. However, its wild resources have been overexploited, leading to market demand far exceeding natural regeneration capacity. Although the Yunnan variety of *Paris polyphylla* has been widely cultivated, it takes 5 to 8 years from seedling emergence to medicinal use due to its perennial nature, resulting in a slow rate of resource regeneration. Additionally, the biosynthesis of steroidal compounds in *Paris polyphylla* is vital resource in the future. However, the synthesis method for Polyphyllin II is much more complex than that for PP10, and the glycosyltransferase responsible for its synthesis remains unidentified¹⁵. These factors suggest that the availability of Polyphyllin II may be limited in the drug development.

Secondly, during our literature review of polyphyllin II, we found that it exhibited cytotoxicity against both HepaRG (liver cancer) and HL-7702 (normal human liver) cells. Specifically, the IC₅₀ values for polyphyllin II were 4.53, 3.259, and 2.465 μM in HepaRG cells after 24, 48, and 72 hours of treatment, respectively, and 1.869, 0.867, and 0.719 μM in HL-7702 cells after the same time points¹⁶. This indicates that Polyphyllin II may have significant hepatotoxicity, which significantly hindered the possibility of drug.. Given the potential safety concerns and the extensive existing

research on the tumor cytotoxicity of Polyphyllin II, we decided to exclude CL6 from further investigation in our study. **We have explained those reasons in manuscript.**

Q2: A major concern is the lack of functional validation of the selective CD133-dependent activities of the studied drugs, PP10 and PP24. As natural compounds often show complex effects on multiple cellular targets, the observed activities may be independent of their binding to CD133. However, there are no mechanistic experiments to support that the described antitumor properties and different effects on cell signaling, mitochondrial physiology, and cell death are caused by blocking CD133 activity. Several related comments that should be addressed are listed below:

1) Cellular thermal shift assay (CETSA) experiments represent crucial biological evidence of direct binding of the drugs to CD133. However, the results and analysis presented are not technically sound: (i) Loading appears uneven, as 45° C samples from both HCT116 and DLD1 show less CD133 in the presence of PP24 than in respective controls. This either indicates that there was a significant technical variance in loading or that the drug decreased CD133 stability.

Answer: Thank you for your insightful comments. We have carefully reviewed the issues you raised, particularly the uneven loading observed in the 45°C samples from both HCT116 and DLD1 cell lines in the presence of PP24. To address this, we have repeated the CETSA assay with improved technical controls and optimizations to ensure more consistent loading and reliable results.

Upon re-evaluation, we confirmed that with the increase of temperature, the CD133 was more stable after treated with PP24 compared with control group (Fig. 6). This finding aligns with our hypothesis and strengthens the biological evidence supporting the interaction of PP24 with CD133. We have incorporated these revised experimental results into the manuscript Fig. 2i, j to provide a more accurate and technically robust assessment of the binding affinity and stability effects of PP24 on CD133.

2) Barely detectable bands in the blots translate only into approx. 2-fold decrease in densitometry (e.g., PP10/60 °C in HCT116 - Fig. 2i vs Fig. 2k; PP10 samples in DLD1 - Fig. 2i vs. Extended Data Fig. 2c).

Answer: Thank you for raising these important points. We have carefully examined the possible reasons for the differences in density observed at 60°C in HCT116 cells, especially for PP10-treated samples. We have identified the density differences in CETSA experiments partly due to background variations. In our initial experiments, the two groups of samples were loaded onto separate gels, which likely resulted in slightly different exposure conditions and background signals. To address this, we've optimized the loading method by placing both groups onto the same gel, ensuring uniform exposure conditions and background. Additionally, to facilitate more accurate analysis and reduce batch-to-batch variations, we have implemented a data normalization process. Each density value obtained from the experiments is now divided by the density value of the control group at 45°C. This normalization step allows for more consistent and reliable comparison of results across different experimental conditions, the result can be seen in Fig. 2 k, I and Extended Data Fig. 2c, d of manuscript.

3) In contrast with these technical discrepancies, error bars in all related graphs show extremely low SD, which is very uncommon for immunoblotting methods. Providing the blots from the two other biological replicates in Extended Data Fig. 2 would strengthen this CETSA analysis and the conclusions made.

Answer: Thanks for your suggestions. To strengthen our CETSA analysis, we have refined the experimental conditions, conducted additional experiments, and updated the results and statistics in Fig. 2 and Extended Data Fig. 2 in the manuscript. We hope these improvements address the concerns regarding the technical aspects of our immunoblotting data.

Fig. 6 (a, b) CETSA results showing the thermal stability of CD133 in the presence of PP10 and PP24. (c, d) Quantitative analysis of the level of CD133 for a, b, normalized by Control-45°C.

Q3: Expression of CD133 was not characterized and compared among the cell lines used for drug validation (Extended Data Fig. 2), which prevents interpretation of CD133-specific cytotoxicity of the drugs. Both HCCLM3 and A549 are known to express CD133, but they showed significantly lower sensitivity to PP10 and PP24. Conversely, CD133 has been previously found absent or minimally expressed in DLD1 (PMID: 19738050; PMID: 22895640; PMID: 26002465), yet DLD1 cells showed similarly high sensitivity to PP10 and PP24 as HTC116 CD133-overexpressing cells. To validate PP10 and PP24 as potential drugs for CD133-targeted therapies, it is necessary to characterize CD133 protein expression and include CRC models with high as well as low/absent CD133 levels to test for the drug selectivity against CD133-overexpressing cells. Otherwise, the observed effects could be attributed to other off-target activities of PP10 and PP24.

Answer: We are thankful for your valuable suggestion. To address the issue of CD133-specific cytotoxicity of PP10 and PP24, we collected various cell lines, including normal colon cell line FHC, CRC cell lines DLD1 and HCT116, lung cancer cell line A549, and liver cancer cell line HCCLM3. We evaluated the CD133 expression levels in these cell lines and found that FHC, A549, and HCCLM3 expressed lower CD133 compared to DLD1 and HCT116 (Fig. 7). We also noted that some CRC cells with high CD133 expression, such as HT29, may be more sensitive to PP10 and PP24 (PMID: 19738050; PMID: 22895640; PMID: 26002465). In these studies, they also compared expression of CD133 among CRC cell lines, we also compared HCT116, DLD1 with HCCLM3 and A549, the expression of CD133 is higher in HCT116 and DLD1 than that in HCCLM3 and A549, which leading to stronger cytotoxicity effect in HCT116 and DLD1 cell lines. Moreover, we constructed a CD133 knockdown HCT116 cell line to test the cytotoxicity of PP10

and PP24 on low CD133-expressing cells (Fig. 2). The results showed weaker cytotoxicity in FHC and CD133 knockdown HCT116 cells compared to WT HCT116 and DLD1 cells, indicating that the cytotoxicity of PP10 and PP24 is related to CD133 expression levels (Fig. 3).

The CD133 expression levels in the cell lines used for drug validation have been characterized and compared. Our findings suggest that the cytotoxicity of PP10 and PP24 correlates with CD133 expression levels. These results could strengthen the evidence for PP10 and PP24 as potential drugs for CD133-targeted therapies.

Fig. 7 Expression level of CD133 in various cell lines. (a) Western blot result of CD133 expression in FHC, DLD1, HCT116, A549, and HCCLM3. (b) Quantitative analysis of the level of CD133 for a.

Q3: The authors performed several experiments to elucidate the mechanism of antitumor effects of PP10 and PP24. However, these experiments lack essential controls - either using CD133 knockdown or interaction mutants (e.g., p85-binding deficient Y828F CD133) - to determine whether removing the target or preventing its interaction with signaling molecules diminishes the effects of PP10/PP24. Without this evidence, it is speculative to claim that the observed changes in cell signaling and other processes are due to the binding of the drugs to CD133 ("Our results fully demonstrate that PP10 and PP24 can specifically bind to CD133 to achieve pan-cancer therapy"). In fact, the potent antitumor activity and divergent mechanisms of action of PP10 vs. PP24 could be related to their different effects on other molecules independent of CD133.

Answer: We are thankful for your meaningful suggestion. To strengthen the link

between PP10 or PP24 interaction with CD133 and their antitumor effects, we constructed a CD133-knockdown HCT116 cell line using shRNA lentiviral vector. After puromycin selection, CD133 expression was dramatically reduced in the shCD133-HCT116 cell line compared to the WT-HCT116 cell line and results can be seen in Fig. 2 of reply. CCK8 assays showed that the IC₅₀ values of PP10 and PP24 in the shCD133-HCT116 and FHC cell lines were over 10 times lower than in the HCT116 cell line, indicating weaker cytotoxicity in CD133-low or normal cells and results can be seen in Fig. 2, 3 of reply. Moreover, we detected the protein expression levels in the PI3K and Wnt pathways for the shCD133-HCT116 cell line and found that knockdown CD133 could inhibit both PI3K/AKT and Wnt/ β -catenin pathway, similar to the anti-tumour effects of PP10 and PP24 treatment. These results suggested that PP10 and PP24 could bind with CD133 to block the downstream pathway and inhibit CRC cells growth.

Q4: Finally, the choice of autophagy inhibitor could flaw the conclusions about PP10-induced autophagy/mitophagy. Chloroquine (CQ) is a suboptimal autophagy inhibitor with known Golgi-related effects, which can paradoxically induce autophagy (PMID: 29940786). This warrants caution when interpreting results obtained by blocking autophagy with this drug. To measure autophagy flux, bafilomycin A1 should be used to inhibit autophagosome-lysosome fusion and cargo sequestration (PMID: 33634751). Importantly, results shown in Fig. 5c, d may indicate that PP10 inhibits rather than promotes autophagy flux. LC3-II forms were accumulated after PP10 treatment alone, and inhibiting the autophagy did not increase their levels, suggesting that PP10 already blocked autophagy in terminal stages (see also 2b in PMID: 33634751). However, using CQ, which clearly could not inhibit autophagy in the case of DLD1 untreated cells (Fig. 5d, lane #1 vs #4), prevents any interpretation of the results. Similarly, microscopy analysis of mitophagy induction cannot be properly interpreted without comparing PP10-treated cells in the presence and absence of autophagy

inhibition by bafilomycin A1. The accumulation of mitochondria in lysosomes may, in fact, again indicate inhibited autophagy at the final stages by PP10 (not increased autophagy). Confocal microscopy results should then be supported by quantitative image analysis across multiple fields of view per biological replicate.

Answer: Thank you for your suggestion. To gain deeper insights into PP10-induced autophagy/mitophagy, we used Bafilomycin A1 (Baf-A1) to inhibit late stage autophagy and assess autophagy flux through Western blot.

Results showed that LC3B-II levels rose with increasing PP10 concentration (lane #1 to #3), and Baf-A1 significantly increased LC3B-II levels compared to the control group (lane #1 vs #4). When compared to the Baf-A1 alone group (lane #4), the combine group slightly increased LC3B-II expression (lane #5), suggesting PP10 may both promote autophagosome formation initially and block their degradation, thereby hindering autophagy at the terminal stage. Similar findings were observed in the P62 group, further indicating autophagosome accumulation and autophagy flux blockage. Finally, the damaged mitochondria could not be removed and accumulated in cell resulting death.

Fig. 8 Expression level of protein within mitophagy pathway. (a, b) Western blot analysis of mitophagy-related proteins in HCT116 and DLD1 cells treated with PP10 and Baf-A1.

If there are any additional suggestions or specific points that require further attention, please do not hesitate to let us know. Your expertise is highly regarded, and we are

eager to address any further recommendations to ensure the excellence of our research.

Thank you for your dedication and commitment to advancing scientific knowledge. We are honored to have our work considered for publication in EMBO Molecular Medicine.

Yours sincerely,

Shaohua Ma, Ph.D.

Institute of Biopharmaceutical and Health Engineering,

Shenzhen International Graduate School,

Tsinghua University, Shenzhen, Guangdong 518055, China

Email: ma.shaohua@sz.tsinghua.edu.cn

Reference

1. Brandes, N., Ofer, D., Peleg, Y., Rappoport, N., and Linial, M. (2022). ProteinBERT: a universal deep-learning model of protein sequence and function. *Bioinformatics* *38*, 2102-2110. 10.1093/bioinformatics/btac020.
2. Xi, G.F., Li, Y.D., Grahovac, G., Rajaram, V., Wadhvani, N., Pundy, T., Mania-Farnell, B., James, C.D., and Tomita, T. (2017). Targeting CD133 improves chemotherapeutic efficacy of recurrent pediatric pilocytic astrocytoma following prolonged chemotherapy. *Molecular Cancer* *16*, 21. 10.1186/s12943-017-0593-z.
3. Jamal, S.M.E., Alamodi, A., Wahl, R.U., Grada, Z., Shareef, M.A., Hassan, S.-Y., Murad, F., Hassan, S.-L., Santourlidis, S., Gomez, C.R., et al. (2020). Melanoma stem cell maintenance and chemo-resistance are mediated by CD133 signal to PI3K-dependent pathways. *Oncogene* *39*, 5468-5478. 10.1038/s41388-020-1373-6.
4. Wei, Y., Jiang, Y., Zou, F., Liu, Y., Wang, S., Xu, N., Xu, W., Cui, C., Xing, Y., Liu, Y.,

- et al. (2013). Activation of PI3K/Akt pathway by CD133-p85 interaction promotes tumorigenic capacity of glioma stem cells. *Proceedings of the National Academy of Sciences of the United States of America* *110*, 6829-6834. 10.1073/pnas.1217002110.
5. Jang, J., Song, Y., Kim, S.-H., Kim, J.-S., Kim, K., Choi, E., Kim, J., and Seo, H. (2017). CD133 confers cancer stem-like cell properties by stabilizing EGFR-AKT signaling in hepatocellular carcinoma. *Cancer letters* *389*, 1-10. 10.1016/j.canlet.2016.12.023.
 6. Sharma, M., Chuang, W., and Sun, Z. (2002). Phosphatidylinositol 3-Kinase/Akt Stimulates Androgen Pathway through GSK3 β Inhibition and Nuclear β -Catenin Accumulation*. *The Journal of Biological Chemistry* *277*, 30935-30941. 10.1074/JBC.M201919200.
 7. Yuan, N., Wang, L., Xi, Q., Zou, N., Zhang, X., Lu, X., and Zhang, Z. (2022). ITGA7, CD133, ALDH1 are inter-correlated, and linked with poor differentiation, lymph node metastasis as well as worse survival in surgical cervical cancer. *The Journal of Obstetrics and Gynaecology Research* *48*, 1011-1018. 10.1111/jog.15163.
 8. Ding, Q., Miyazaki, Y., Tsukasa, K., Matsubara, S., Yoshimitsu, M., and Takao, S. (2014). CD133 facilitates epithelial-mesenchymal transition through interaction with the ERK pathway in pancreatic cancer metastasis. *Mol Cancer* *13*, 15. 10.1186/1476-4598-13-15.
 9. Ren, Q., Chen, J., and Liu, Y. (2021). LRP5 and LRP6 in Wnt Signaling: Similarity and Divergence. *Frontiers in Cell and Developmental Biology* *9*. 10.3389/fcell.2021.670960.

10. Wu, J., Bai, Y., Lu, Y., Yu, Z., Zhang, S., Yu, B., Chen, L., and Li, J. (2024). Role of sex steroids in colorectal cancer: pathomechanisms and medical applications. *Am J Cancer Res* *14*, 3200-3221. 10.62347/oeps6893.
11. Topi, G., Ghatak, S., Satapathy, S., Ehrnström, R., Lydrup, M., and Sjölander, A. (2022). Combined Estrogen Alpha and Beta Receptor Expression Has a Prognostic Significance for Colorectal Cancer Patients. *Frontiers in Medicine* *9*. 10.3389/fmed.2022.739620.
12. Sato, T., Oshi, M., Huang, J.L., Chida, K., Roy, A., Endo, I., and Takabe, K. (2024). CD133 expression is associated with less DNA repair, better response to chemotherapy and survival in ER-positive/HER2-negative breast cancer. *Breast cancer research and treatment*. 10.1007/s10549-024-07434-3.
13. Kouzmenko, A., Takeyama, K., Ito, S., Furutani, T., Sawatsubashi, S., Maki, A., Suzuki, E., Kawasaki, Y., Akiyama, T., Tabata, T., and Kato, S. (2004). Wnt/ β -Catenin and Estrogen Signaling Converge in Vivo*. *Journal of Biological Chemistry* *279*, 40255-40258. 10.1074/JBC.C400331200.
14. Floch, L., Rivat, C., De Wever, O., Bruyneel, E., Mareel, M., Dale, T., and Gespach, C. (2005). The proinvasive activity of Wnt-2 is mediated through a noncanonical Wnt pathway coupled to GSK-3 β and c- Jun/AP-1 signaling. *The FASEB Journal* *19*. 10.1096/fj.04-2373fje.
15. Hua, X., Kou, C., Wang, F., Zhang, J., Yuan, J., and Xue, Z. (2025). Steroidal compounds in *Paris polyphylla*: structure, biological activities, and biosynthesis. *Curr Opin Plant Biol* *84*, 102695. 10.1016/j.pbi.2025.102695.

16. Wang, W., Dong, X., You, L., Sai, N., Leng, X., Yang, C., Yin, X., and Ni, J. (2018). Apoptosis in HepaRG and HL-7702 cells induced by polyphyllin II through caspases activation and cell-cycle arrest. *Journal of Cellular Physiology* 234, 7078-7089. [10.1002/jcp.27462](https://doi.org/10.1002/jcp.27462).

17th Jul 2025

Dear Dr. Ma,

Thank you for the submission of your revised manuscript to EMBO Molecular Medicine. We have now heard back from the two referees who agreed to re-evaluate your manuscript. As you will see from the reports below, while both referees acknowledge the improvements in the revised manuscript, they also remain critical regarding the limited mechanistic insight. After a consultation with my colleagues here, we concluded that raised concerns are justified and should be addressed in an additional and final round of major revision. Providing deeper mechanistic insight is essential for further consideration of the manuscript.

Please also amend following points:

1) Authors: Please provide an institutional email address for the co-corresponding author Yubo Zhang in the manuscript and our submission system. Also, make sure that email addresses for all authors are correct in our submission system.

2) Figures:

- We note that in Extended Data Fig. 12 lung control image and lung PP24 5mg/kg image are identical. Please check, provide an explanation and correct.

- Please upload 7 Extended Data Figures as Expanded View (EV) Figures. EV Figures should be uploaded as individual high-resolution files with their legends placed in the main manuscript file after the main figure legends. Please rename the figures to Figure EV1 etc. and update their callouts in the main manuscript text. Please check "Author Guidelines" for more information: <https://www.embopress.org/page/journal/17574684/authorguide#figureformat>

<https://www.embopress.org/page/journal/17574684/authorguide#expandedview>

- The rest of Extended Data Figures should be compiled in Appendix and uploaded as a single PDF file. Please rename the figures to Appendix Figure S1, etc. and update their callouts in the main manuscript text. Appendix should have a title page with a table of contents and page numbers.

- Please remove all figures from the manuscript file and leave only their legends at the end of the file.

3) In the main manuscript file, please do the following:

- Please address all comments suggested by our data editors listed below:

o Figure legends:

1. Please indicate what */ **/ ***/ **** represents; if this represents p value(s), please indicate the statistical test used and where appropriate, specify the exact p value in the legend(s) of figure(s) 1B, C, F, H.

2. Please note that the exact p values are not provided in the legends of figures 2K, L; 3D, E; 6G, 7D, E, G, I; extended data figure(s) 2C, D; 3C, D, G, I; 4C, E, G, I; 5A-D; 7A, B, C, D, H; 8A, E, F.

3. Please indicate the statistical test used for data analysis in the legends of figures 2K, L; 3D, E; 4A, C, D, E, F; 6A, B, G; 7D, E, G, I; extended data figure(s) 2C, D; 3C, D, G, I; 4C, E, G, I; 5A-D; 7A, B, C, D, H; 8A, E, F.

4. Please note that information related to n is missing in the legends of figures 3A, 6A, G; extended data figure(s) 9A-C.

5. Please note that the error bars are not defined in the legends of extended data figure(s) 9A-C.

6. Please note that the scale bar is missing for figure 5F.

7. Please note that scale bar and its definition are missing for figures 5A, E.

- Please add a callout for the figure 7I and make sure that the figures are called out sequentially.

- Rename "Summary" to "Abstract".

- In Methods, provide the antibody dilutions that were used for each antibody.

- Indicate in legends exact n and exact p values, not a range, along with the statistical test used. To keep the figures "clear"

some authors found providing an Appendix table Sx with all exact p-values preferable. You are welcome to do this if you want to.

- Please provide Reagents and Tools Table and uploaded it as a separate file. Structured Methods section includes Reagents and Tools Table followed by a Methods and Protocols section. More information on how to adhere to this format as well as downloadable templates (.docx) for the Reagents and Tools Table can be found in our author guidelines:

<https://www.embopress.org/page/journal/17574684/authorguide#structuredmethods>

An example of a paper with Structured Methods can be found here:

<https://www.embopress.org/doi/full/10.1038/s44320-024-00037-6#sec-4>

- Rename "Competing interests" to "Disclosure and competing interests statement". We updated our journal's competing interests policy in January 2022 and request authors to consider both actual and perceived competing interests. Please review the policy <https://www.embopress.org/competing-interests> and update your competing interests if necessary.

- Author contributions: Please remove it from the manuscript. CRedit has replaced the traditional author contributions section because it offers a systematic machine-readable author contributions format that allows for more effective research assessment. You are encouraged to use the free text boxes beneath each contributing author's name to add specific details on the author's contribution. More information is available in our guide to authors:

<https://www.embopress.org/page/journal/17574684/authorguide#authorshippinguidelines>

- Data availability: Please use the following format to report the accession number of your data:

[data type]: [full name of the resource] [accession number/identifier] [(doi or URL or identifiers.org/DATABASE:ACCESSION)]

Please check "Author Guidelines" for more information.

<https://www.embopress.org/page/journal/17574684/authorguide#availabilityofpublishedmaterial>

- Please correct the reference citation in the text and reference list. In the text a reference should be cited by author and year of publication. Include a space between a word and the opening parenthesis of the reference that follows. In the reference list, citations should be listed in alphabetical order. Where there are more than 10 authors on a paper, 10 will be listed, followed by "et al.". Also, please remove DOIs. DOIs should only be used for preprints and datasets that have not been published. Please check "Author Guidelines" for more information.

<https://www.embopress.org/page/journal/17574684/authorguide#referencesformat>

4) Funding: Please make sure that information about all sources of funding are complete in both our submission system and in the manuscript. Currently, National Natural Science Foundation of China (32371470 and 82341019); Merck Research Grant, and the Cross-disciplinary Research and Innovation Fund of Tsinghua SIGS (No. JC2022007); Guangdong Basic and Applied Basic Research Foundation (2023B1515120025); Shenzhen Fundamental Research Program (No. JCYJ20240813112004006); Shenzhen Major Science and Technology Projects (KJZD20230923115400001) are missing in our submission system.

5) The Paper Explained: Please provide "The Paper Explained" and add it to the main manuscript text. Please check "Author Guidelines" for more information. <https://www.embopress.org/page/journal/17574684/authorguide#researcharticleguide>

6) Synopsis: Every published paper now includes a 'Synopsis' to further enhance discoverability. Synopses are displayed on the journal webpage and are freely accessible to all readers. They include separate synopsis image and synopsis text.

- Synopsis image: Please provide a visual abstract as a high-resolution jpeg file 550 px-wide x 300-600 pixels high to illustrate your article.

- Synopsis text: Please provide a short standfirst (maximum of 300 characters, including space) as well as 2-5 one sentence bullet points that summarise the paper as a .doc file. Please write the bullet points to summarise the key NEW findings. They should be designed to be complementary to the abstract - i.e. not repeat the same text. We encourage inclusion of key acronyms and quantitative information (maximum of 30 words / bullet point). Please use the passive voice.

Further consideration of a revision that addresses reviewer's concerns in full will entail an additional round of review.

Acceptance or rejection of the manuscript will depend on the completeness of your responses included in the next, final version of the manuscript. For this reason, and to save you from any frustrations in the end, I would strongly advise against returning an incomplete revision.

We would welcome the submission of a revised version within three months for further consideration. Please let us know if you require longer to complete the revision.

I look forward to receiving your revised manuscript.

Yours sincerely,

Zeljko Durdevic

Zeljko Durdevic
Senior Editor
EMBO Molecular Medicine

We require:

2) Individual production quality figure files as .eps, .tif, .jpg (one file per figure). For guidance, download the 'Figure Guide PDF': (<https://www.embopress.org/page/journal/17574684/authorguide#figureformat>).

3) A .docx formatted letter INCLUDING the reviewers' reports and your detailed point-by-point responses to their comments. As part of the EMBO Press transparent editorial process, the point-by-point response is part of the Review Process File (RPF), which will be published alongside your paper.

4) A complete author checklist, which you can download from our author guidelines (<https://www.embopress.org/page/journal/17574684/authorguide#submissionofrevisions>). Please insert information in the checklist that is also reflected in the manuscript. The completed author checklist will also be part of the RPF.

6) It is mandatory to include a 'Data Availability' section after the Materials and Methods. Before submitting your revision, primary datasets produced in this study need to be deposited in an appropriate public database, and the accession numbers and database listed under 'Data Availability'. Please remember to provide a reviewer password if the datasets are not yet public (see <https://www.embopress.org/page/journal/17574684/authorguide#dataavailability>).

12) Author contributions: You will be asked to provide CRediT (Contributor Role Taxonomy) terms in the submission system. These replace a narrative author contribution section in the manuscript.

13) A Conflict of Interest statement should be provided in the main text.

14) Every published paper now includes a 'Synopsis' to further enhance discoverability. Synopses are displayed on the journal webpage and are freely accessible to all readers. They include a short stand first (maximum of 300 characters, including space) as well as 2-5 one-sentences bullet points that summarizes the paper. Please write the bullet points to summarize the key NEW findings. They should be designed to be complementary to the abstract - i.e. not repeat the same text. We encourage inclusion of key acronyms and quantitative information (maximum of 30 words / bullet point). Please use the passive voice. Please attach these in a separate file or send them by email, we will incorporate them accordingly.

15) Include a Reagents and Tools Table as part of the Methods section, which can be downloaded from our author guidelines (<https://www.embopress.org/page/journal/17574684/authorguide#structuredmethods>)

***** Reviewer's comments *****

Referee #1 (Comments on Novelty/Model System for Author):

As stated last time, the topic is overall interesting, although the work focuses on applying the existing AI tools, but not really developing some new tools, to facilitate the identification of new compounds. After the revision, there are some information added to improve the work though still not very extensive in terms of revealing new mechanisms. So I would maintain my view that this work is of medium novelty/impact.

Referee #1 (Remarks for Author):

The authors have addressed my concerns for Q1-4. Meanwhile, the KEGG analysis for the DLD1 cell line in Q5 was not provided.

Referee #2 (Remarks for Author):

This is an improved version, clearly showing the authors' effort to address potential weaknesses of their work. However, the revision appears a bit rushed, as many of the important additional experiments and analyses were not included in the manuscript. I particularly do not understand the authors' decision not to present the results with shCD133 cells, which would significantly strengthen their conclusions. Although the authors' point-by-point response is partially satisfying (see the remaining issues listed below), the revised manuscript, as it is presented, still lacks mechanistic biological evidence to support the validity of the AI-based drug discovery approach.

Remaining issues preventing the correct interpretation of the results or undermining the conclusions made:

1) CETSA analysis has been repeated, but the design of the analysis is still flawed. The authors do not provide any evidence of the same loading, i.e., same CD133 levels under normal conditions (w/o heating) in both control and treated groups. Samples without any heating should be used for normalization and interpretation of the thermal shift, as is standard in the field (e.g., see Fig. 2 in <https://doi.org/10.1016/j.xpro.2022.101423>). Indeed, for a proper comparison, it is crucial to run all samples (treated vs untreated) on the same gel/membrane. However, by reviewing the provided source data, there are at most duplicates for PP10 or PP24 CETSA analysis conducted in this design, and none contain loading/normalization controls. Without determining the CD133 levels at normal (control) temperature, it remains speculative whether the thermal shift is caused by the drug binding or

by differences in basal CD133 expression between groups.

2) Experiments in CD133-high vs. low. cell lines and in CD133-knockdown cells are important to support drug selectivity. The authors provided some of this evidence in their response, but did not incorporate this data (addressing comments of both reviewers) into their manuscript.

3) Similarly, to support the PP10-induced autophagy/mitophagy, the authors performed experiments using the suggested bafilomycin A1, but did not include this data in the manuscript. Bafilomycin A1 is a gold standard inhibitor to study autophagy flux, and it is important to include these results to support any conclusions. Importantly, the experiments presented in the authors' response Fig. 8 were not performed correctly. To determine how PP10 affects the autophagy flux, it is crucial to use the same concentration of the autophagy inhibitor for untreated and treated cells. However, the authors used a 30-fold higher concentration of the inhibitor for treated cells, which itself may easily lead to more efficient autophagy inhibition and increased accumulation of LC3-II forms.

Referee #1 (Comments on Novelty/Model System for Author):

As stated last time, the topic is overall interesting, although the work focuses on applying the existing AI tools, but not really developing some new tools, to facilitate the identification of new compounds. After the revision, there are some information added to improve the work though still not very extensive in terms of revealing new mechanisms. So I would maintain my view that this work is of medium novelty/impact.

Referee #1 (Remarks for Author):

Q1: The authors have addressed my concerns for Q1-4. Meanwhile, the KEGG analysis for the DLD1 cell line in Q5 was not provided.

Answer: Thank you for raising this point. We apologize for not including the KEGG analysis for the DLD1 cell line in our previous response to Q5. Below is the KEGG result for PP24: In the KEGG analysis of PP24-treated DLD1 cells, the Wnt signaling pathway ranks significantly lower than other pathways, and the total count of identified pathways is relatively low. Given these results, we have moved this analysis to Fig. EV5G rather than including it in the main figures.

Fig 2 KEGG pathway enrichment analysis showing the functional pathways affected by DEGs in DLD1 cells following PP24 treatment.

Referee #2 (Remarks for Author):

This is an improved version, clearly showing the authors' effort to address potential weaknesses of their work. However, the revision appears a bit rushed, as many of the important additional experiments and analyses were not included in the manuscript. I particularly do not understand the authors' decision not to present the results with shCD133 cells, which would significantly strengthen their conclusions. Although the authors' point-by-point response is partially satisfying (see the remaining issues listed below), the revised manuscript, as it is presented, still lacks mechanistic biological evidence to support

the validity of the AI-based drug discovery approach.

Remaining issues preventing the correct interpretation of the results or undermining the conclusions made:

Q1: CETSA analysis has been repeated, but the design of the analysis is still flawed. The authors do not provide any evidence of the same loading, i.e., same CD133 levels under normal conditions (w/o heating) in both control and treated groups. Samples without any heating should be used for normalization and interpretation of the thermal shift, as is standard in the field (e.g., see Fig. 2 in <https://doi.org/10.1016/j.xpro.2022.101423>). Indeed, for a proper comparison, it is crucial to run all samples (treated vs untreated) on the same gel/membrane. However, by reviewing the provided source data, there are at most duplicates for PP10 or PP24 CETSA analysis conducted in this design, and none contain loading/normalization controls. Without determining the CD133 levels at normal (control) temperature, it remains speculative whether the thermal shift is caused by the drug binding or by differences in basal CD133 expression between groups.

Answer: Thanks for your advice, we used a slightly different method to perform CETSA analysis(1-3). Unlike the study mentioned, which compared wild-type and mutated proteins from different cell lines using a Flag tag for loading control, we used the same cell line (HCT116 or DLD1) for each experiment. In our study, HCT116 and DLD1 cells were collected and lysed using liquid nitrogen. The cell lysate was split into two equal parts—one treated with PP10 or PP24, and the other with DMSO as a control. After incubation at room temperature for an hour, samples were exposed to different temperatures (45 °C, 50 °C, 55 °C, 60 °C) for 3 minutes each, then chilled on ice for 5 minutes. Post-centrifugation, supernatants were collected and CD133 expression was detected via Western blotting (Fig. 3).

To ensure comparable CD133 levels between control and treated groups, we used the same cell line for each experiment and divided the cell lysate into equal portions. This ensured equal CD133 input for both groups. Additionally, the loading volume was kept the same during Western blotting. The control and drug-treated groups were loaded onto the same gel and membrane to standardize exposure conditions and background, the indication of membrane can be seen. It could ensure that any observed thermal shift was due to drug binding rather than differences in CD133 expression (Fig. 4).

Fig 3 Schematic diagram of the experimental method of CESTA

Fig 4 Indication diagram of the WB membrane of CESTA

Q2: Experiments in CD133-high vs. low. cell lines and in CD133-knockdown cells are important to support drug selectivity. The authors provided some of this evidence in their response, but did not incorporate this data (addressing comments of both reviewers) into their manuscript.

Answer: We deeply appreciate the insightful suggestion from the reviewer. In response, we have incorporated the CD133 - knockdown data, along with pathway changes and cell viability test results, into Figure 2K, L and Figure EV1B - E of the manuscript. Additionally, we have added explanations of these results to the text.

Q3: Similarly, to support the PP10-induced autophagy/mitophagy, the authors performed experiments using the suggested bafilomycin A1, but did not include this data in the manuscript. Bafilomycin A1 is a gold standard inhibitor to study autophagy flux, and it is important to include these results to support any conclusions. Importantly, the experiments presented in the authors' response Fig. 8 were not performed correctly. To determine how PP10 affects the autophagy

flux, it is crucial to use the same concentration of the autophagy inhibitor for untreated and treated cells. However, the authors used a 30-fold higher concentration of the inhibitor for treated cells, which itself may easily lead to more efficient autophagy inhibition and increased accumulation of LC3-II forms.

Answer: We are grateful for your suggestion. We have included the data from the Bafilomycin A1 experiments in the manuscript, which can be found in Fig. EV4C, and we have also added related explanations in the manuscript and highlighted them. We sincerely apologize for the mistake in the Baf-A1 concentration in #5. We mistakenly pasted the concentration of PP10 directly as the concentration of Baf-A1 without making the necessary correction. It should be 0.1 μM , consistent with that in #4. We have corrected this error in both the response and the manuscript (Fig. 5).

Fig 5 Expression level of protein within mitophagy pathway. (a, b) Western blot analysis of mitophagy-related proteins in HCT116 and DLD1 cells treated with PP10 and Baf-A1.

If there are any additional suggestions or specific points that require further attention, please do not hesitate to let us know. Your expertise is highly regarded, and we are eager to address any further recommendations to ensure the excellence of our research.

Thank you for your dedication and commitment to advancing scientific knowledge. We are honored to have our work considered for publication in EMBO Molecular Medicine.

Yours sincerely,

Shaohua Ma, Ph.D.

Institute of Biopharmaceutical and Health Engineering,

Shenzhen International Graduate School,

Tsinghua University, Shenzhen, Guangdong 518055, China

Email: ma.shaohua@sz.tsinghua.edu.cn

Reference

1. Jafari R, Almqvist H, Axelsson H, Ignatushchenko M, Lundbäck T, Nordlund P, et al. The cellular thermal shift assay for evaluating drug target interactions in cells. *Nat Protoc.* 2014;9(9):2100-22.
2. Ding XJ, Cai XM, Wang QQ, Liu N, Zhong WL, Xi XN, et al. Vitexicarpin suppresses malignant progression of colorectal cancer through affecting c-Myc ubiquitination by targeting IMPDH2. *Phytomedicine.* 2024;132:155833.
3. Xia J, Xu M, Hu H, Zhang Q, Yu D, Cai M, et al. 5,7,4'-Trimethoxyflavone triggers cancer cell PD-L1 ubiquitin-proteasome degradation and facilitates antitumor immunity by targeting HRD1. *MedComm (2020).* 2024;5(7):e611.

28th Aug 2025

Dear Dr. Ma,

We have now heard back from the one referee who agreed to evaluate your manuscript. This referee also assessed author responses to concerns raised by referee #2. We are pleased to inform you that your manuscript is accepted for publication and is now being sent to our publisher to be included in the next available issue of EMBO Molecular Medicine.

Zeljko Durdevic
Senior Editor
EMBO Molecular Medicine

Referee #1 (Remarks for Author):

This reviewer does not have further comments.
